# LEARNING HIERARCHICAL PROTEIN REPRESENTATIONS VIA COMPLETE 3D GRAPH NETWORKS

**Limei Wang[1]\*, Haoran Liu[1]\*, Yi Liu[2]\*†, Jerry Kurtin[1], Shuiwang Ji[1]†**

[1] Texas A&M University, College Station, TX
[2] Florida State University, Tallahassee, FL
`{limei, liuhr99, jkurtin, sji}@tamu.edu, liuy@cs.fsu.edu`

## ABSTRACT

We consider representation learning for proteins with 3D structures. We build 3D graphs based on protein structures and develop graph networks to learn their representations. Depending on the levels of details that we wish to capture, protein representations can be computed at different levels, *e.g.*, the amino acid, backbone, or all-atom levels. Importantly, there exist hierarchical relations among different levels. In this work, we propose to develop a novel hierarchical graph network, known as ProNet, to capture the relations. Our ProNet is very flexible and can be used to compute protein representations at different levels of granularity. By treating each amino acid as a node in graph modeling as well as harnessing the inherent hierarchies, our ProNet is more effective and efficient than existing methods. We also show that, given a base 3D graph network that is complete, our ProNet representations are also complete at all levels. Experimental results show that ProNet outperforms recent methods on most datasets. In addition, results indicate that different downstream tasks may require representations at different levels. Our code is publicly available as part of the DIG library (`https://github.com/divelab/DIG`).

## 1 INTRODUCTION

Proteins consist of one or more amino acid chains and perform various functions by folding into 3D conformations. Learning representations of proteins with 3D structures is crucial for a wide range of tasks (Cao et al., 2021; Strokach et al., 2020; Wu et al., 2021; Yang et al., 2019; Ganea et al., 2022; Stärk et al., 2022; Morehead et al., 2022a;b; Liu et al., 2020). In machine learning, molecules, proteins, etc. are usually modeled as graphs (Liu et al., 2022; Fout et al., 2017; Jumper et al., 2021; Gao et al., 2021; Gao & Ji, 2019; Yan et al., 2022; Wang et al., 2022b; Yu et al., 2022; Xie et al., 2022a;b; Gui et al., 2022; Luo et al., 2022). With the advances of deep learning, 3D graph neural networks (GNNs) have been developed to learn from 3D graph data (Liu et al., 2022; Jumper et al., 2021; Xie & Grossman, 2018; Liu et al., 2021; Joshi et al., 2023). In this work, we build 3D graphs based on protein structures and develop 3D GNNs to learn protein representations.

Depending on the levels of granularity we wish to capture, we construct protein graphs at different levels, including the amino acid, backbone, and all-atom levels, as shown in Fig. 1. Specifically, each node in constructed graphs represents an amino acid, and each amino acid possesses internal structures at different levels. Importantly, there exist hierarchical relations among different levels. Existing methods for protein representation learning either ignore hierarchical relations within proteins (Jing et al., 2021b; Zhang et al., 2023), or suffer from excessive computational complexity (Jing et al., 2021a; Hermosilla et al., 2021) as shown in Table 1. In this work, we propose a novel hierarchical graph network, known as ProNet, to learn protein representations at different levels. Our ProNet effectively captures the hierarchical relations naturally present in proteins.

By constructing representations at different levels, our ProNet effectively integrates the inherent hierarchical relations of proteins, resulting in a more rational protein learning scheme. Building on a novel hierarchical fashion, our method can achieve great efficiency, even at the most complex

---

\*Equal contributions
†Equal senior contributions

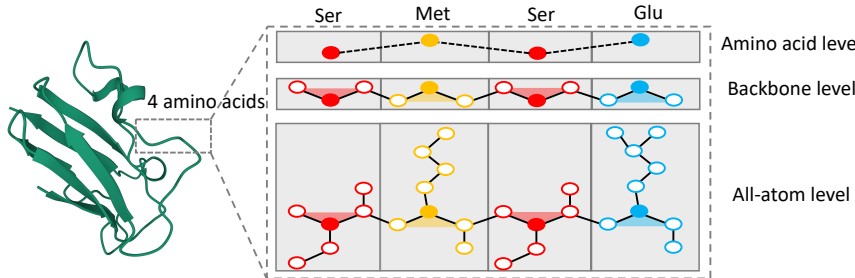

Figure 1: Illustration of hierarchical representations of proteins. Different colors indicate different types of amino acids. The filled circles are $C_\alpha$ atoms, and non-filled are other atoms. Each amino acid has different levels of inner structures. From coarse-grained to fine-grained levels, we can use $C_\alpha$ coordinates, backbone atom coordinates, or all-atom coordinates to represent the protein structure. *Note that we treat each amino acid as a node in the graph modeling despite different levels.* The actual atoms are in 3D, and this illustration uses 2D for simplicity.

all-atom level. In addition, completeness at all levels enable models to generate informative and discriminative representations. Practically, ProNet possesses great flexibility for different data and downstream tasks. Users can easily choose the level of granularity at which the model should operate based on their data and tasks. We conduct experiments on multiple downstream tasks, including protein fold and function prediction, protein-ligand binding affinity prediction, and protein-protein interaction prediction. Results show that ProNet outperforms recent methods on most datasets. We also show that different data and tasks may require representations at different levels.

## 2 BACKGROUND

Representation learning of small molecules with 3D structures has been studied recently (Schütt et al., 2017; Klicpera et al., 2020; Liu et al., 2022; Wang et al., 2022a), and existing methods can fully determine 3D structures of molecules (Wang et al., 2022a). However, representation learning of proteins with 3D structures is still challenging due to the large number of atoms and special hierarchies that naturally present in protein structures. Existing methods for protein representation learning either ignore hierarchical relations within proteins, or suffer from excessive computational complexity, as explained in Table 1. Detailed related work is listed in Sec. 5. In this section, we first introduce hierarchical structures of proteins in Sec. 2.1, which inspires us to design hierarchical representations of proteins in Sec. 3. We then introduce existing complete 3D graph networks in Sec. 2.2, which can be used to capture protein structures completely.

### 2.1 HIERARCHICAL PROTEIN STRUCTURES

Proteins are macromolecules consisting of one or more chains of amino acids. Each chain may contain up to hundreds or even thousands of amino acids. An amino acid consists of an amino (-NH$_2$) group, a carboxyl (-COOH) group, and a side chain that is unique to each amino acid. The functional groups are all attached to the alpha carbon ($C_\alpha$) atom. The $C_\alpha$ atoms, together with the corresponding amino group and carboxyl group, form the backbone of a protein. As shown in Fig. 1, we can use $C_\alpha$ coordinates, backbone atom coordinates, or all-atom coordinates to represent protein structures, leading to three levels of representations. Note that protein structures are traditionally organized into primary, secondary, tertiary, and quaternary levels, and our categorization of levels is different. Next, we can use complete 3D graph networks to fully capture protein structures at three levels.

Table 1: Comparisons of existing protein learning methods. Firstly, treating atoms instead of amino acids as **nodes** leads to high **complexity**. Here $n$, $N$, and $k$ denote the number of amino acids, the number of atoms, and the average degree in a 3D protein graph, and $N \gg n$. In addition, most existing methods only capture one **level** of protein structures, and only IEConv considers **hierarchical relations** of proteins using several pooling layers. Our method can learn hierarchical representations at three levels. Lastly, our method can capture 3D structures **completely** at all levels.

| Method | Node | Complexity | Hierarchical Level | Hierarchical Relations | Complete or not |
|---|---|---|---|---|---|
| GearNet (Zhang et al., 2023) | Amino Acid | $O(nk)$ | Amino Acid | ✗ | ✗ |
| GVP-GNN etc. (Jing et al., 2021b; Ingraham et al., 2019) | Amino Acid | $O(nk)$ | Backbone | ✗ | ✓ |
| vector-gated GVP-GNN (Jing et al., 2021a) | Atom | $O(Nk)$ | All-Atom | ✗ | ✓ |
| IEConv (Hermosilla et al., 2021) | Atom | $O(Nk)$ | All-Atom | ✓ | ✗ |
| **Ours** | Amino Acid | $O(nk)$ | Amino Acid $\to$ Backbone $\to$ All-Atom | ✓ | ✓ ✓ ✓ |

## 2.2 COMPLETE 3D GRAPH NETWORKS

**3D Graphs.** Many real-world data can be modeled as 3D graphs. A 3D graph can be represented as $G = (\mathcal{V}, \mathcal{E}, \mathcal{P})$. Here, $\mathcal{V} = \{\mathbf{v}_i\}_{i=1,\dots,n}$ is the set of node features, where each $\mathbf{v}_i \in \mathbb{R}^{d_v}$ denotes the feature vector for node $i$. $\mathcal{E} = \{\mathbf{e}_{ij}\}_{i,j=1,\dots,n}$ is the set of edge features, where $\mathbf{e}_{ij} \in \mathbb{R}^{d_e}$ denotes the edge feature vector for edge $ij$. $\mathcal{P} = \{P_i\}_{i=1,\dots,n}$ is the set of position matrices, where $P_i \in \mathbb{R}^{k_i \times 3}$ denotes the position matrix for node $i$. $k_i$ can be different for different applications. For example, if we treat each atom in a molecule as a node, then $k_i = 1$ for each node $i$. For a protein, if we treat each amino acid as a node, then $k_i$ is the number of atoms in amino acid $i$. In our method, we represent proteins as 3D graphs and learn hierarchical representations of proteins in Sec. 3.

**Complete Geometric Representations.** As defined in ComENet (Wang et al., 2022a), a geometric transformation $\mathcal{F}(\cdot)$ is complete if for two 3D graphs $G^1 = (\mathcal{V}, \mathcal{E}, \mathcal{P}^1)$ and $G^2 = (\mathcal{V}, \mathcal{E}, \mathcal{P}^2)$, the geometric representations $\mathcal{F}(G^1) = \mathcal{F}(G^2) \iff \exists R \in \text{SE}(3)$, for $i = 1, \dots, n$, $P_i^1 = R(P_i^2)$. Here SE(3) is the Special Euclidean group that includes all rotations and translations in 3D. Based on the definition, ComENet proposes a complete representation for small molecules with torsion angles and spherical coordinates.

**Complete Message Passing Scheme.** By incorporating complete geometric representations to the commonly-used message passing framework (Gilmer et al., 2017), we achieve a complete message passing scheme as

$$\mathbf{v}_i^{l+1} = \text{UPDATE}\left(\mathbf{v}_i^l, \sum_{j \in \mathcal{N}_i} \text{MESSAGE}\left(\mathbf{v}_j^l, \mathbf{e}_{ji}, \mathcal{F}(G)\right)\right), \tag{1}$$

where $\mathcal{N}_i$ denotes the set of node $i$'s neighbors, and UPDATE and MASSAGE functions are usually implemented by neural networks or mathematical operations.

## 3 HIERARCHICAL REPRESENTATIONS OF PROTEIN STRUCTURES

**Notations.** To learn representations of proteins with 3D structures, we first model a protein as a 3D graph $G = (\mathcal{V}, \mathcal{E}, \mathcal{P})$ as introduced in Sec. 2.2. Specifically, we treat each amino acid as a node and define edges between nodes using a cutoff radius. That is, if the distance between two nodes is less than a predefined radius, there is an edge between these two nodes. For a node $i$, the node feature $\mathbf{v}_i$ is the one-hot embedding of the amino acid type. For an edge $ij$, the edge feature $\mathbf{e}_{ij}$ is an embedding of the sequential distance $j - i$, following existing studies (Ingraham et al., 2019; Zhang et al., 2023). In addition, the position matrix $P_i$ for a node $i$ includes the coordinates of all atoms in the amino acid if available. Note that the rows in $P_i$ are given in a fixed atom order. For example, for the amino acid alanine, the atom order in the position matrix is $N$, $C_\alpha$, $C$, $O$, and $C_\beta$.

Considering the hierarchical structures of amino acids and proteins as introduced in Sec. 2.1, we propose to learn protein representations at different levels, including the amino acid, backbone, and all-atom levels, as shown in Fig. 1. In addition, we aim to capture protein structures completely at each level. Therefore, for levels from top to bottom in Fig. 1, we design complete geometric representations as $\mathcal{F}(G)_{\text{base}}$, $\mathcal{F}(G)_{\text{bb}}$, and $\mathcal{F}(G)_{\text{all}}$, respectively. By incorporating the designed complete geometric representations into Eq. 1, we can fully capture protein structures at all levels.

### 3.1 AMINO ACID LEVEL REPRESENTATIONS

At the amino acid level, we treat each amino acid as a node and use $C_\alpha$ coordinates as the position of the node. This leads to the most coarse-grained representation of the protein. By ignoring the detailed inner structures like the backbone and side chain of amino acids, methods designed for small molecules can be applied directly to learn amino acid level representations of proteins. To achieve complete representations, we design the geometric representation $\mathcal{F}(G)_{\text{base}}$ at the amino acid level as $\{(d_{ji}, \theta_{ji}, \phi_{ji}, \tau_{ji})\}_{i=1,\dots,n,\ j \in \mathcal{N}_i}$ following ComENet (Wang et al., 2022a). Here $(d_{ji}, \theta_{ji}, \phi_{ji})$ is the spherical coordinate of node $j$ in the local coordinate system of node $i$ to determine the relative position of $j$, and $\tau_{ji}$ is the rotation angle of edge $ji$ to capture the remaining degree of freedom.

ComENet is used to obtain complete representations at the amino acid level in our study. However, **our method is significantly different from ComENet**. Firstly and most importantly, we study protein representation learning based on the unique structural properties of proteins. As a result, we

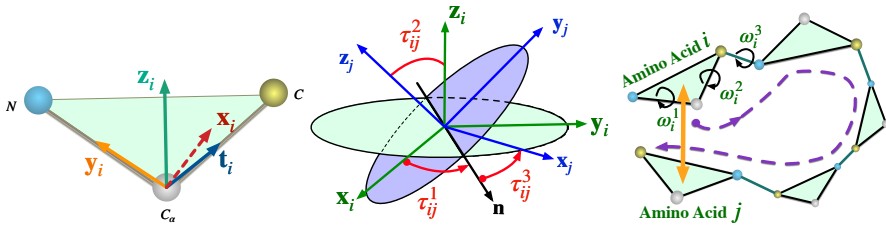

(a)        (b)        (c)

Figure 2: Illustrations of our proposed backbone-level representations. (a) Construction of the backbone coordinate system for amino acid $i$. (b) Computation of the three Euler angles between the backbone coordinate systems for two amino acids $i$ and $j$. (c) Illustrations of determining the relative rotation between amino acids $i$ and $j$ for existing methods and our proposed method. The purple dashed line indicates how existing methods determine the relative rotation between $i$ and $j$ by computing all backbone dihedral angles along the chain. The yellow arrow shows how our method determines the relative rotation between amino acids $i$ and $j$ using only three Euler angles.

design a hierarchical protein learning framework, which can incorporate the inherent hierarchies in protein structures and can largely advance protein representation learning. ComENet is designed for small molecules whose structures are less complicated than protein structures. Secondly, we seek to obtain complete representations at each hierarchical level, and ComENet can only be applied to learn complete representations at the amino acid level. Given $\mathcal{F}(G)_{\text{base}}$ at the amino acid level, we further design $\mathcal{F}(G)_{\text{bb}}$ and $\mathcal{F}(G)_{\text{all}}$ based on the unique structural properties of proteins to learn complete representations at all levels. Thirdly and technically, even at the amino acid level, we use a different strategy to define the local coordinate system (LCS) for each amino acid, not directly analogizing amino acids in our methods to atoms in ComENet. The LCS is used to compute $(d_{ji}, \theta_{ji}, \phi_{ji}, \tau_{ji})$ for each edge $ji$. Our strategy is also developed based on protein structures. Specifically, we define the LCS for node $i$ based on nodes $i-1$ and $i+1$ following existing protein learning studies (Ingraham et al., 2019). ComENet defines it based on $i$-th nearest neighbor $f_i$ and second nearest neighbor $s_i$, which requires extra computation to sort the neighbors and find the nearest two. Therefore, our method can reduce the computation of ComENet and is more efficient.

### 3.2    BACKBONE LEVEL REPRESENTATIONS WITH EULER ANGLES

Building on the proposed amino acid level representation, we further consider all backbone atoms for each amino acid to derive finer-grained protein representations. Since we can fully capture $C_\alpha$ coordinates via the complete geometric representation $\mathcal{F}(G)_{\text{base}}$ at the amino acid level, the remaining degree of freedom at the backbone level is the rotation between two backbone planes. This is because, with such rotation, we can easily determine the coordinates of other backbone atoms besides $C_\alpha$ atoms based on rigid bond lengths and bond angles (Jumper et al., 2021). Therefore, we propose to use Euler angles to capture such rotation. Specifically, we first define the local coordinate system for an amino acid $i$ as $\mathbf{y}_i = \mathbf{r}_i^N - \mathbf{r}_i^{C_\alpha}$, $\mathbf{t}_i = \mathbf{r}_i^C - \mathbf{r}_i^{C_\alpha}$, $\mathbf{z}_i = \mathbf{t}_i \times \mathbf{y}_i$, and $\mathbf{x}_i = \mathbf{y}_i \times \mathbf{z}_i$, as shown in Fig. 2(a). We then compute three Euler angles $\tau_{ij}^1$, $\tau_{ij}^2$, and $\tau_{ij}^3$ between two backbone coordinate systems as shown in Fig. 2(b). Here, $\mathbf{n} = \mathbf{z}_i \times \mathbf{z}_j$ is the intersection of two local system, $\tau_{ij}^1$ is the signed angle between $\mathbf{n}$ and $\mathbf{x}_i$, $\tau_{ij}^2$ is the angle between $\mathbf{z}_i$ and $\mathbf{z}_j$, and $\tau_{ij}^3$ is the angle from $\mathbf{n}$ to $\mathbf{x}_j$. By considering these three Euler angles, the orientation between any two backbone planes can be determined, thereby fully capturing backbone structures of proteins. Thus, the complete geometric representation at this level is $\mathcal{F}(G)_{\text{bb}} = \mathcal{F}(G)_{\text{base}} \cup \{(\tau_{ji}^1, \tau_{ji}^2, \tau_{ji}^3)\}_{i=1,\dots,n,\, j \in \mathcal{N}_i}$.

**Advantages of using Euler angles.** Most existing approaches directly integrate backbone information into amino acid features. Specifically, they compute three backbone dihedral angles $\omega_i^1$, $\omega_i^2$, and $\omega_i^3$ (Ingraham et al., 2019; Jing et al., 2021a; Li et al., 2022) for each amino acid $i$ based on $N_i$, $C_{\alpha_i}$, $C_i$, and $N_{i+1}$ atoms as shown in Fig. 2(c). Then the sin and cos values for the three angles are part of the node features of amino acid $i$. For any two amino acids $i$ and $j$, if we safely assume $j > i$, the relative rotation of these two backbone triangles is determined by all the amino acids between $i$ and $j$ along the protein chain. Thus, the relative rotation is determined by all the $(j - i + 1) \times 3$ bond rotation angles $\{\omega_k^1, \omega_k^2, \omega_k^3\}_{k=i,\dots,j}$, as shown in Fig. 2(c). However, our proposed backbone level method can determine the relative rotation for any two amino acids $i$ and $j$ by only three Euler angles $\tau_{ji}^1$, $\tau_{ji}^2$, and $\tau_{ji}^3$, regardless of the sequential distance $j - i$ along the protein chain. Hence, our method can significantly improve the efficiency of representation learning at this level.

### 3.3 ALL-ATOM LEVEL REPRESENTATIONS WITH SIDE CHAIN TORSION ANGLES

To obtain the most fine-grained representations of proteins, we consider all atoms in each amino acid. As introduced in Sec. 2.1, an amino acid consists of backbone atoms and side chain atoms. Therefore, building on our backbone level representation, we further incorporate side chain information, leading to the all-atom level representation as shown in Fig. 1. We assume all bond lengths and bond angles in each amino acid are fully rigid (Jumper et al., 2021), then the degree of freedom we need to consider is torsion angles in side chains (Jumper et al., 2021). There are at most five torsion angles for any amino acid. For example, as shown in Fig. 5 in Appendix A, the alanine has zero side chain torsion angle, the cysteine has only one, and the leucine has two. Note that only the amino acid arginine has five side chain torsion angles, and the fifth angle is close to 0. Therefore, we only consider the first four torsion angles for efficiency, denoted as $\chi^1, \chi^2, \chi^3, \chi^4$. We list the atoms used to compute side chain torsion angles for each amino acid in Table 7 in Appendix A. With such side chain torsion angles, we can determine the side chain structure for each amino acid. Based on the backbone level representation, the geometric representation at this level is $\mathcal{F}(G)_{\text{all}} = \mathcal{F}(G)_{\text{bb}} \cup \{(\chi_i^1, \chi_i^2, \chi_i^3, \chi_i^4)\}_{i=1,\dots,n}$. Note that although side chain torsion angles are important properties of protein structures, it is only used in recent studies (Jumper et al., 2021) for all-atom coordinates prediction, and none of the existing protein representation learning methods use it to capture protein structure information. Here we incorporate it for protein representation learning, leading to our all-atom level representation.

**Differences with existing all-atom level methods.** Several existing studies also consider all-atom information of proteins (Hermosilla et al., 2021; Jing et al., 2021a), but our method is significantly different and possesses unique advantages, as illustrated in

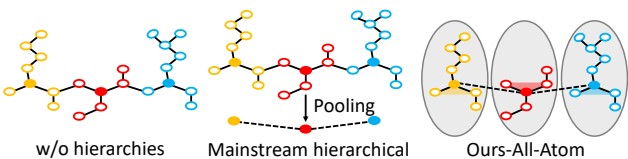

w/o hierarchies    Mainstream hierarchical    Ours-All-Atom

Figure 3: Illustrations of three kind of all-atom level methods.

Fig. 3. Specifically, The vector-gated GVP-GNN (Jing et al., 2021a) belongs to the *"w/o hierarchies"* methods in Fig. 3. It treats each atom as a node and uses an equivariant GNN to update node features. However, the important hierarchical information is not considered. IEConv (Hermosilla et al., 2021) follows the *"mainstream hierarchical"* methods in Fig. 3. It treats each atom as a node and employs several pooling layers to obtain representations at different levels. For each level, it needs several intrinsic-extrinsic convolution layers for message passing. By treating an atom as a node and employing a large model with more than ten layers, IEConv induces excessive computing costs. Our proposed method ***effectively preserves hierarchical relations of proteins.*** In addition, ***by treating each amino acid as a node and integrating side chain torsion angles as node features, our method has much fewer nodes in constructed graphs,*** resulting in a much more efficient learning scheme. We provide mathematical expressions and explanations in Table 1. We also conduct experiments in Sec. 6.5 to show the efficiency and effectiveness of our all-atom level method.

### 3.4 COMPLETENESS ANALYSIS

In this study, our primary objective is to develop a hierarchical representation learning framework for proteins based on their unique structural properties. After constructing a 3D graph $G = (\mathcal{V}, \mathcal{E}, \mathcal{P})$ for a protein from the framework, we further achieve complete representations at three hierarchical levels. Intuitively, complete representations can capture all details of 3D protein structures and enable our method to generate distinct representations for different 3D graphs, subject to rigid transformations such as rotation and translation. Based on the definition of completeness in Sec. 2.2 and Wang et al. (2022a), we rigorously show how our method can achieve completeness at all three levels. We also summarize whether existing methods are complete or not in Table 1.

**Completeness of the Amino Acid Level.** At the amino acid level, we only consider $C_\alpha$ coordinate of each amino acid. The methods designed for small molecules can be applied directly at this level. As shown in Sec. 3.1, we design our geometric representation $\mathcal{F}(G)_{\text{base}}$ of a 3D protein graph based on ComENet (Wang et al., 2022a). Since ComENet is provably complete, and the different definition of a local coordinate system in our method does not affect the completeness proof of ComENet, our method can naturally achieve completeness at this level.

**Completeness of the Backbone Level.** Given a complete $\mathcal{F}(G)_{\text{base}}$, we rigorously show the geometric representation at the backbone level $\mathcal{F}(G)_{\text{bb}} = \mathcal{F}(G)_{\text{base}} \cup \{(\tau_{ji}^1, \tau_{ji}^2, \tau_{ji}^3)\}_{i=1,\dots,n,\, j \in \mathcal{N}_i}$ is

complete in Appendix B.1. Note that our proof is based on the assumption that all bond lengths and bond angles are fully rigid in amino acids, and this assumption is widely accepted (Jumper et al., 2021). Intuitively, a complete geometric representation at the backbone level can capture all 3D information of the backbone structure. As protein backbones largely determine protein functions (Lopez & Mohiuddin, 2020; Nelson et al., 2008), capturing fine details of them can benefit various tasks.

**Completeness of the All-Atom Level.** Given a complete $\mathcal{F}(G)_{\text{bb}}$, we rigorously show the geometric representation at the all-atom level $\mathcal{F}(G)_{\text{all}} = \mathcal{F}(G)_{\text{bb}} \cup \{(\chi_i^1, \chi_i^2, \chi_i^3, \chi_i^4)\}_{i=1,...,n}$ is complete in Appendix B.2. With the complete geometric representation at this level, our method can fully capture 3D information of all atoms in a protein. Therefore, our method can distinguish any two distinct protein structures in nature. Especially, our all-atom method can capture side chain structures compared with the backbone level method. Side chains are important for proteins (Spassov et al., 2007). The tertiary and quaternary structures of a protein are determined by interactions between side chains and environment (O'Connor et al., 2010). In addition, interactions between side chains also play a crucial role in protein-protein and protein-ligand interactions (Tanaka & Scheraga, 1976; Berka et al., 2009). Overall, the all-atom method can capture information for both inter- and intra-protein interactions, leading to better performances on various tasks.

## 4 PRONET

Based on the message passing scheme Eq. 1 and the hierarchical geometric representations in Sec. 3, we propose our ProNet for hierarchical protein representation learning as shown in Fig. 4. The inputs to ProNet are node features, edge features, and geometric representations. Following existing graph neural networks (Schütt et al., 2017; Wang et al., 2022a), our ProNet contains several

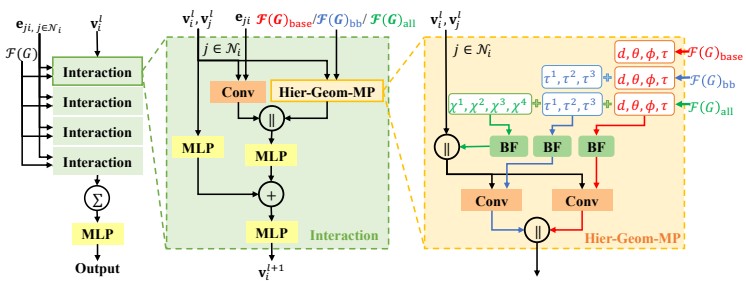

Figure 4: An illustration of ProNet. $\|$ denotes concatenation. Conv denotes a graph convolution layer to update node features. Hier-Geom-MP denotes the proposed hierarchical message passing layer. BF denotes basis functions to embed distances and angles. Details of the model architecture and basis functions are provided in Appendix C.2.

*interaction blocks* to update node features and one *Readout function* to obtain graph-level representations. The Readout function includes a summation function and several fully-connected layers. Specifically, in the interaction block, we design our novel hierarchical message passing layer *Hier-Geom-MP* to learn protein representations based on node features, edge features, and either one level of geometric representations. Hier-Geom-MP is specially designed for protein learning and can effectively capture hierarchical relations of proteins. The *Conv* is adapted from GraphConv (Morris et al., 2019) and is used to update node features based on edge features. A detailed description of the model architecture is provided in Appendix C. Note that there are three levels of geometric representations, and our model only takes one as input. The three levels of geometric representations result in three levels of ProNet, namely ProNet-Amino Acid, ProNet-Backbone, and ProNet-All-Atom. Users can easily adapt the framework to different downstream tasks by specifying the level of geometric representations. ***Our experiment results in Sec. 6 also show that different data and tasks may require representations at different levels.***

## 5 RELATED WORK

Learning protein representations is essential to a variety of tasks in protein engineering. Existing methods for protein learning consider different kinds of protein information, including amino acid sequences (Öztürk et al., 2018; Bepler & Berger, 2019; Rao et al., 2019; Elnaggar et al., 2021; Bileschi et al., 2022), protein surfaces (Gainza et al., 2020; Sverrisson et al., 2021; Dai & Bailey-Kellogg, 2021; Somnath et al., 2021), and protein 3D structures (Fout et al., 2017; Gligorijević et al., 2021; Baldassarre et al., 2021; Jing et al., 2021a; Cha et al., 2022). Due to recent advances in protein structure prediction (Senior et al., 2020; Jumper et al., 2021; Varadi et al., 2022; Baek et al., 2021),

structures of many proteins are becoming available with high accuracy. In addition, protein structures are crucial for protein functions. In this work, we focus on representation learning of proteins with 3D structures. Earlier studies formulate proteins as 3D grid-like data and employ 3D CNNs for learning (Derevyanko et al., 2018; Townshend et al., 2021). However, the grid-like data is extremely sparse, leading to expensive learning cost and unsatisfactory performance. Hence, recent studies model proteins as 3D graphs and use 3D GNNs to learn representations (Hermosilla et al., 2021; Hermosilla & Ropinski, 2022; Jing et al., 2021b; Ingraham et al., 2019; Zhang et al., 2023; Li et al., 2022). Based on the analysis in Sec. 3 and Table 1, previous methods on 3D protein graphs can be categorized into three levels, including amino acid level, backbone level, and all-atom level. For example, GearNet (Zhang et al., 2023) treats an amino acid as a node and uses amino acid types as node features, thus it is categorized as an amino acid level method. GVP-GNN (Jing et al., 2021b) represents protein backbone structures with backbone dihedral angles computed from backbone atoms. Thus it is a backbone level method. IEConv (Hermosilla et al., 2021) treats each atom as a node in protein graphs, therefore, it is an all-atom level method. ***The differences between existing methods and our ProNet are illustrated mathematically in Table 1 and explained in details at the end of Sec. 3.1, Sec. 3.2, and Sec. 3.3.***

## 6 EXPERIMENTS

We evaluate our ProNet on various protein tasks, including protein fold and reaction prediction, protein-ligand binding affinity prediction, and protein-protein interaction prediction. Detailed descriptions of the datasets are provided in Appendix D. We also conduct ablation study on the design of our all-atom method in Sec. 6.5, showing the efficiency and effectiveness of our method. Detailed experimental setup and optimal hyperparameters are provided in Appendix E. Additional experimental results are provided in Appendix F. The code is integrated in the 3Dgraph part of DIG library (Liu et al., 2021) and available at `https://github.com/divelab/DIG`.

### 6.1 FOLD CLASSIFICATION

Protein fold classification (Hou et al., 2018; Levitt & Chothia, 1976) is crucial to capture protein structure-function relations and protein evolution. Following the dataset and experimental settings in Hou et al. (2018) and Hermosilla et al. (2021), we evaluate our methods on the fold classification task. A detailed description of the data is provided in Appendix D. In total, this dataset contains 16,712 proteins from 1,195 folds. There are three test sets, namely Fold, Superfamily, and Family. We report the accuracies on the three test sets and the average

Table 2: Accuracy (%) on fold and reaction classification tasks. The top two results are highlighted as **1st** and 2nd.

| Method | React | Fold | | | |
|---|---|---|---|---|---|
| | | Fold | Sup. | Fam. | Avg. |
| GCN (Kipf & Welling, 2017) | 67.3 | 16.8 | 21.3 | 82.8 | 40.3 |
| DeepSF (Hou et al., 2018) | 70.9 | 17.0 | 31.0 | 77.0 | 41.7 |
| GVP-GNN (Jing et al., 2021b) | 65.5 | 16.0 | 22.5 | 83.8 | 40.8 |
| IEConv (Hermosilla et al., 2021) | **87.2** | 45.0 | 69.7 | 98.9 | 71.2 |
| New IEConv (Hermosilla & Ropinski, 2022) | **87.2** | 47.6 | 70.2 | 99.2 | 72.3 |
| HoloProt (Somnath et al., 2021) | 78.9 | – | – | – | – |
| DWNN (Li et al., 2022) | 76.7 | 31.8 | 37.8 | 85.2 | 51.5 |
| GearNet (Zhang et al., 2023) | 79.4 | 28.4 | 42.6 | 95.3 | 55.4 |
| GearNet-IEConv (Zhang et al., 2023) | 83.7 | 42.3 | 64.1 | 99.1 | 68.5 |
| GearNet-Edge (Zhang et al., 2023) | 86.6 | 44.0 | 66.7 | 99.1 | 69.9 |
| GearNet-Edge-IEConv (Zhang et al., 2023) | 85.3 | 48.3 | **70.3** | **99.5** | 72.7 |
| ProNet-Amino Acid | 86.0 | 51.5 | 69.9 | 99.0 | 73.5 |
| ProNet-Backbone | 86.4 | **52.7** | **70.3** | 99.3 | **74.1** |
| ProNet-All-Atom | 85.6 | 52.1 | 69.0 | 99.0 | 73.4 |

of the three accuracy values in Table 2. The results for baseline methods are taken from original papers (Hermosilla et al., 2021; Somnath et al., 2021; Zhang et al., 2023; Li et al., 2022).

Table 2 shows that our methods can achieve the best results on two of the three test sets and the best average value. For Superfamily and Family, our methods outperform all of the baseline methods and achieve similar performance as GearNet-Edge-IEConv. But GearNet-Edge-IEConv uses edge message passing scheme, which is more computationally expensive than the node message passing

Table 3: Comparisons between ProNet and other methods in terms of computational cost on the Fold dataset using the same Nvidia GeForce RTX 2080 Ti 11GB GPU.

| Method | Hierarchical Level | Time (sec.) | | Converge Time |
|---|---|---|---|---|
| | | Train | Inference | |
| GearNet-Edge (Zhang et al., 2023) | Amino Acid | **OOM** | – | – |
| GearNet-Edge-IEConv (Zhang et al., 2023) | Amino Acid | **OOM** | – | – |
| GVP-GNN (Jing et al., 2021b) | Backbone | 35 | 6 | ∼ 9 h |
| IEConv (Hermosilla et al., 2021) | All-Atom | 165 | 22 | ∼ 24 h |
| ProNet-Amino Acid | Amino Acid | 32 | 5 | ∼ 9 h |
| ProNet-Backbone | Backbone | 32 | 6 | ∼ 9 h |
| ProNet-All-Atom | All-Atom | 32 | 6 | ∼ 9 h |

scheme in our method, as discussed in Liu et al. (2022). In addition, as shown in Table 3, GearNet-Edge-IEConv can not be trained using one Nvidia GeForce RTX 2080 Ti 11GB GPU due to its high complexity. For Fold, the most difficult one among the three test sets, all of our methods on three levels can significantly outperform baseline methods, and ProNet-backbone improves the accuracy from 48.3% to 52.7%, demonstrating the good generalization ability of our methods. Our methods also set the new state of the art for the average value.

## 6.2   REACTION CLASSIFICATION

Enzymes are proteins that act as biological catalysts. They can be classified with enzyme commission (EC) numbers which groups enzymes based on the reactions they catalyze (Webb, 1992; Omelchenko et al., 2010). We follow the dataset and experiment settings in Hermosilla et al. (2021) to evaluate our methods on this task. In total, this dataset contains 37,428 proteins from 384 EC numbers (Berman et al., 2000; Dana et al., 2019). Comparison results are summarized in Table 2, where the results for baseline methods are taken from original papers (Hermosilla et al., 2021; Hermosilla & Ropinski, 2022; Li et al., 2022; Zhang et al., 2023). Our methods achieve better or comparable results compared with previous methods. Note that IEConv methods (Hermosilla et al., 2021; Hermosilla & Ropinski, 2022) are complicated and have larger numbers of parameters. Specifically, the numbers of parameters for IEConv methods are about 10M and 20M, while that of our methods are less than 2M.

## 6.3   LIGAND BINDING AFFINITY

Computational prediction of protein-ligand binding affinity (LBA) is essential for many downstream tasks in drug discovery as it mitigates the cost of wet-lab experiments and accelerates virtual screening (Huang et al., 2021). In this task, we use the dataset curated from PDBbind (Wang et al., 2004; Liu et al., 2015) and experiment settings in Somnath et al. (2021). We adopt dataset split with 30% and 60% sequence identity thresholds to verify the generalization ability of our models for unseen proteins. In terms of experiment settings, we employ the two-branch network following Somnath et al. (2021) for fair comparison. We use the same ligand network as Holoprot (Somnath et al., 2021) and use our ProNet as the protein network. Detailed experimental setup is provided in Appendix E.

Comparison results are summarized in Table 4, where the baseline results are taken from Somnath et al. (2021) and Townshend et al. (2021). Results are reported for 3 experimental runs. The detailed standard deviation of experiment results are provided in Appendix F. Note that methods in Atom3D use a different experiment setting than other methods. Therefore, it is not fair to compare our results with Atom3D methods. However, we still include their results in Table 4 in case readers are interested in their setting. Specifically, the models in Atom3D are trained with binding pockets only, making the task less challenging. This is because the binding affinity would be highly related to binding structure (Lu et al., 2022), the models that take binding pockets as input incorporate prior information on the binding site, binding pose, and the interaction between protein and ligand. Other baseline methods do not consider such prior information in the input. The results show that our methods achieve either best or second best results on both splits and obtain significantly better results than previous state-of-the-art methods on the sequence identity 60% split. For our methods at different

Table 4: Results on protein-ligand binding affinity prediction task. The top two results are highlighted as **1st** and 2nd. * denotes methods trained with the complex binding pockets only, which provides prior information on the interaction between protein and ligand and makes the task less challenging.

| Method | Sequence Identity 30% | | | Sequence Identity 60% | | |
|---|---|---|---|---|---|---|
| | RMSE ↓ | Pearson ↑ | Spearman ↑ | RMSE ↓ | Pearson ↑ | Spearman ↑ |
| Atom3D-3DCNN* (Townshend et al., 2021) | **1.416** | 0.550 | **0.553** | 1.621 | 0.608 | 0.615 |
| Atom3D-ENN* (Townshend et al., 2021) | 1.568 | 0.389 | 0.408 | 1.620 | 0.623 | 0.633 |
| Atom3D-GNN* (Townshend et al., 2021) | 1.601 | 0.545 | 0.533 | 1.408 | 0.743 | 0.743 |
| DeepDTA (Öztürk et al., 2018) | 1.866 | 0.472 | 0.471 | 1.762 | 0.666 | 0.663 |
| Bepler and Berger (2019) (Bepler & Berger, 2019) | 1.985 | 0.165 | 0.152 | 1.891 | 0.249 | 0.275 |
| TAPE (Rao et al., 2019) | 1.890 | 0.338 | 0.286 | 1.633 | 0.568 | 0.571 |
| ProtTrans (Elnaggar et al., 2021) | 1.544 | 0.438 | 0.434 | 1.641 | 0.595 | 0.588 |
| MaSIF (Gainza et al., 2020) | 1.484 | 0.467 | 0.455 | 1.426 | 0.709 | 0.701 |
| IEConv (Hermosilla et al., 2021) | 1.554 | 0.414 | 0.428 | 1.473 | 0.667 | 0.675 |
| Holoprot-Full Surface (Somnath et al., 2021) | 1.464 | 0.509 | 0.500 | 1.365 | 0.749 | 0.742 |
| Holoprot-Superpixel (Somnath et al., 2021) | 1.491 | 0.491 | 0.482 | 1.416 | 0.724 | 0.715 |
| ProNet-Amino Acid | 1.455 | 0.536 | 0.526 | 1.397 | 0.741 | 0.734 |
| ProNet-Backbone | 1.458 | 0.546 | 0.550 | 1.349 | 0.764 | 0.759 |
| ProNet-All-Atom | 1.463 | **0.551** | 0.551 | **1.343** | **0.765** | **0.761** |

levels, the all-atom one is best on 5 out of 6 metrics. As the binding affinity may correlate to the chemical reactions on the side chain of a protein, the results may imply that the all-atom method can capture more information for both inter- and intra- protein interaction.

## 6.4 PROTEIN PROTEIN INTERACTION

Protein-protein interactions (PPI) are involved in most cellular processes and essential for biological applications (Ganea et al., 2022). For example, antibody proteins bind to antigens to recognize diseases (Townshend et al., 2021). Following the dataset (Townshend et al., 2019; Vreven et al., 2015) and experiment settings in Townshend et al. (2021), we predict whether two amino acids contact when the two proteins bind. The evaluation metric is AUROC. Results in Table 5 show that our all-atom level method outperforms all previous methods, improving the result from 0.866 to 0.871.

Table 5: Results on the PPI task. The top two results are highlighted as **1st** and 2nd.

| Method | AUROC ↑ |
| --- | --- |
| Atom3D-3DCNN (Townshend et al., 2021) | 0.844 |
| Atom3D-GNN (Townshend et al., 2021) | 0.669 |
| GVP-GNN (Jing et al., 2021a) | 0.866 |
| ProNet-Amino Acid | 0.857 |
| ProNet-Backbone | 0.858 |
| ProNet-All-Atom | **0.871** |

In addition, the results for three levels may imply that our all-atom representation can capture more details from side chains on both interacting proteins and thus benefits the binding site prediction.

## 6.5 OBSERVATIONS AND ABLATION STUDIES

**Observations: different downstream tasks may require methods at different levels.** As shown in Table 2, *our ProNet-backbone outperforms the methods of the other two levels on function prediction tasks*, including fold and reaction classification tasks. This indicates the backbone-level method can capture details from the folding structure of proteins, rendering better predictions for protein functions. In contrast to function prediction tasks, as shown in Table 4 and Table 5, *our ProNet-all-atom outperforms the methods of the other levels on most metrics of interaction prediction tasks, namely LBA and PPI tasks.* This observation implies that our all-atom level method is able to capture fine-grained side chain structure information, eventually contributing to the predictions of binding affinity and binding sites for interacted proteins.

**Ablation studies on all-atom level.** As discussed in Sec. 3.3, we conduct experiments on three all-atom level methods to show the advantages of our proposed all-atom level method. We adopt the same base model (Wang et al., 2022a) for fair comparison. The results are shown in Table 6. The first baseline is *"w/o hierarchies"*, where each atom is treated as a node in a 3D protein

Table 6: Comparison of three all-atom methods on the Fold dataset. The **best** results are highlighted in the table. All the models are trained using the same computing infrastructure (Nvidia GeForce RTX 2080 TI 11GB) for fair comparison. The training time is the average time per epoch, and the three methods use similar epochs to converge.

| Method | Time (sec.) | | Accuracy (%) | | | |
| --- | --- | --- | --- | --- | --- | --- |
| | Train | Inference | Fold | Sup. | Fam. | Avg. |
| w/o hierarchies | 181.2 | 18.1 | 36.9 | 49.5 | 94.2 | 60.2 |
| Mainstream hierarchical | 148.7 | 17.7 | 51.5 | 68.7 | **99.0** | 73.1 |
| ProNet-All-Atom | **32.1** | **6.3** | **52.1** | **69.0** | **99.0** | **73.4** |

graph. The performance of this method is unsatisfying, possibly due to the lack of hierarchical information in graph modeling. In addition, it takes a much longer time for both training and inference. The second baseline is *"Mainstream hierarchical"* with one hierarchical pooling layer, which naturally requires a two-level architecture. The first level follows the *"w/o hierarchies"* method, and the second level treats each amino acid as a node with features obtained by aggregating representations of atoms in the corresponding amino acid. The computational cost is high since two base models are involved for two levels. Our proposed ProNet-All-Atom achieves the best performance among the three methods with less computing time. Overall, our method is both efficient and effective.

## 7 CONCLUSION

Protein structures are crucial for protein functions and can be represented at different levels, including the amino acid, backbone, and all-atom levels. We propose ProNet to capture hierarchical relations among different levels and learn protein representations. Particularly, ProNet is complete at all levels, leading to informative and discriminative representations. Results show that ProNet outperforms previous methods on most datasets, and different tasks may require representations at different levels.

ACKNOWLEDGMENTS

This work was supported in part by National Science Foundation grant IIS-2006861 and National Institutes of Health grant U01AG070112.

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

# Appendix

## A SIDE CHAIN TORSION ANGLES

Figure 5: Illustration of amino acid structures. The red circles are side chain torsion angles.

Table 7: Atoms for computing the side chain torsion angles for each amino acid.

|  | $\chi^1$ | $\chi^2$ | $\chi^3$ | $\chi^4$ | $\chi^5$ |
|---|---|---|---|---|---|
| ALA |  |  |  |  |  |
| ARG | $N, C_\alpha, C_\beta, C_\gamma$ | $C_\alpha, C_\beta, C_\gamma, C_\delta$ | $C_\beta, C_\gamma, C_\delta, N_\epsilon$ | $C_\gamma, C_\delta, N_\epsilon, C_\zeta$ | $C_\delta, N_\epsilon, C_\zeta, N_{\eta 1}$ |
| ASN | $N, C_\alpha, C_\beta, C_\gamma$ | $C_\alpha, C_\beta, C_\gamma, O_{\delta 1}$ |  |  |  |
| ASP | $N, C_\alpha, C_\beta, C_\gamma$ | $C_\alpha, C_\beta, C_\gamma, O_{\delta 1}$ |  |  |  |
| CYS | $N, C_\alpha, C_\beta, S_\gamma$ |  |  |  |  |
| GLN | $N, C_\alpha, C_\beta, C_\gamma$ | $C_\alpha, C_\beta, C_\gamma, C_\delta$ | $C_\beta, C_\gamma, C_\delta, O_{\epsilon 1}$ |  |  |
| GLU | $N, C_\alpha, C_\beta, C_\gamma$ | $C_\alpha, C_\beta, C_\gamma, C_\delta$ | $C_\beta, C_\gamma, C_\delta, O_{\epsilon 1}$ |  |  |
| GLY |  |  |  |  |  |
| HIS | $N, C_\alpha, C_\beta, C_\gamma$ | $C_\alpha, C_\beta, C_\gamma, N_{\delta 1}$ |  |  |  |
| ILE | $N, C_\alpha, C_\beta, C_{\gamma 1}$ | $C_\alpha, C_\beta, C_{\gamma 1}, C_{\delta 1}$ |  |  |  |
| LEU | $N, C_\alpha, C_\beta, C_\gamma$ | $C_\alpha, C_\beta, C_\gamma, C_{\delta 1}$ |  |  |  |
| LYS | $N, C_\alpha, C_\beta, C_\gamma$ | $C_\alpha, C_\beta, C_\gamma, C_\delta$ | $C_\beta, C_\gamma, C_\delta, C_\epsilon$ | $C_\gamma, C_\delta, C_\epsilon, N_\zeta$ |  |
| MET | $N, C_\alpha, C_\beta, C_\gamma$ | $C_\alpha, C_\beta, C_\gamma, S_\delta$ | $C_\beta, C_\gamma, S_\delta, C_\epsilon$ |  |  |
| PHE | $N, C_\alpha, C_\beta, C_\gamma$ | $C_\alpha, C_\beta, C_\gamma, C_{\delta 1}$ |  |  |  |
| PRO | $N, C_\alpha, C_\beta, C_\gamma$ | $C_\alpha, C_\beta, C_\gamma, C_\delta$ |  |  |  |
| SER | $N, C_\alpha, C_\beta, O_\gamma$ |  |  |  |  |
| THR | $N, C_\alpha, C_\beta, O_{\gamma 1}$ |  |  |  |  |
| TRP | $N, C_\alpha, C_\beta, C_\gamma$ | $C_\alpha, C_\beta, C_\gamma, C_{\delta 1}$ |  |  |  |
| TYR | $N, C_\alpha, C_\beta, C_\gamma$ | $C_\alpha, C_\beta, C_\gamma, C_{\delta 1}$ |  |  |  |
| VAL | $N, C_\alpha, C_\beta, C_{\gamma 1}$ |  |  |  |  |

We list atoms used to compute side chain torsion angles for each amino acid in Table 7. Note that AlphaFold2 (Jumper et al., 2021) also considers alternative side chain torsion angles. This is because some side chains parts are $180°$-rotation-symmetric, and the torsion angle $\chi$ and $\chi + \pi$ result in the same physical structure with the internal atom names changed. But in our method, atom names are given, therefore, we do not need to consider the alternative side chain torsion angles.

## B PROOFS

As defined in Wang et al. (2022a) and discussed in Sec. 2.2, the formal definition of completeness for 3D protein graphs is shown as below.

**Definition 1** (Completeness). *For two protein graphs $G^1 = (\mathcal{V}, \mathcal{E}, \mathcal{R}^1)$ and $G^2 = (\mathcal{V}, \mathcal{E}, \mathcal{R}^2)$, where $\mathcal{R}^1 = \{R_i^1\}_{i=1,\dots,n}$ and $\mathcal{R}^2 = \{R_i^2\}_{i=1,\dots,n}$, respectively, a geometric representation $\mathcal{F}(G)$ is considered complete if*

$$\mathcal{F}(G^1) = \mathcal{F}(G^2) \iff \exists \mathcal{T} \in SE(3), \text{ for } i = 1, \dots, n, \; \mathcal{R}_i^1 = \mathcal{T}(\mathcal{R}_i^2). \tag{2}$$

Here SE(3) is the Special Euclidean group that includes all rotations and translations in 3D. To show whether $\mathcal{F}(G)$ is complete, we need to prove in both directions. We first need to show Eq. 2 holds from right to left, which is obvious. This is because our proposed $\mathcal{F}(G)$ is based on relative information such as distances and angles, thus it is naturally SE(3) invariant. Secondly, we need to show that Eq. 2 holds from left to right. Essentially, we need to show that a 3D structure can be uniquely determined by $\mathcal{F}(G)$. In this section, we rigorously show that our proposed $\mathcal{F}(G)_{\text{bb}}$ and $\mathcal{F}(G)_{\text{all}}$ are complete.

### B.1 Proof of the Completeness for Backbone Level Representation

*Proof.* Based on Def. 1, we need to prove that the coordinates of all backbone atoms in each amino acid can be uniquely determined given $\mathcal{F}(G)_{\text{bb}}$. As $\mathcal{F}(G)_{\text{base}}$ is complete for amino acid level, the positions of all $C_\alpha$ atoms in a 3D protein graph are determined as stated in Sec. 3.4. Thus the remaining degree of freedom at the backbone level is the rotation between two backbone planes. This is because, with such rotation, we can easily determine the coordinates of other backbone atoms besides $C_\alpha$ atoms based on the rigid bond lengths and bond angles. Hence, building on the amino acid level, we only need to prove that the local coordinate system for each backbone plane can be uniquely determined.

We prove this by induction. First, we denote $n$ as the number of nodes, *i.e.,* amino acids, in a 3D protein graph. Apparently, the case $n = 1$ holds. Assume the case $n = k$ holds that the geometric representation $\mathcal{F}(G)_{\text{bb}}$ is complete, thus, the locations of all the $k$ backbone planes are uniquely determined. Then we need to prove the proposition holds for the $n = k + 1$ case. Without losing generality, we denote node $j$ as the $(k + 1)$-th node, which is connected to node $i$ among existing $k$ nodes, forming a connected graph $G$ of size $(k + 1)$. We then prove that the local coordinate system of the $(k + 1)$-th backbone plane is uniquely determined by the proposed Euler angles $(\tau_{ji}^1, \tau_{ji}^2, \tau_{ji}^3)$. As illustrated in Fig. 2(b), we use unit vectors $(\mathbf{x}_i, \mathbf{y}_i, \mathbf{z}_i)$ and $(\mathbf{x}_j, \mathbf{y}_j, \mathbf{z}_j)$ to denote the backbone coordinate axes of node $i$ and node $j$, respectively. The intersection vector between plane $\mathbf{x}_i\mathbf{y}_i$ and $\mathbf{x}_j\mathbf{y}_j$ is denoted as $\mathbf{n} = \mathbf{z_i} \times \mathbf{z_j}$. Given the Euler angles $\tau_{ji}^1, \tau_{ji}^2, \tau_{ji}^3$, we have

$$\mathbf{x}_i \cdot \mathbf{n} = \cos \tau_{ji}^1, \tag{3}$$

$$\mathbf{x}_i \times \mathbf{n} \cdot \mathbf{z}_i = \sin \tau_{ji}^1, \tag{4}$$

$$\mathbf{z}_i \cdot \mathbf{z}_j = \cos \tau_{ji}^2, \tag{5}$$

$$\mathbf{n} \cdot \mathbf{x}_j = \cos \tau_{ji}^3, \tag{6}$$

$$\mathbf{n} \times \mathbf{x}_j \cdot \mathbf{z}_j = \sin \tau_{ji}^1. \tag{7}$$

Then we sequentially prove by contradiction that vectors $\mathbf{z}_j$, $\mathbf{x}_j$, and $\mathbf{y}_j$ can be uniquely determined by the Euler angles.

Assume the coordination system for the backbone of node $j$ is not unique, *i.e.*, there are alternative unit vectors $(\mathbf{x}_j', \mathbf{y}_j', \mathbf{z}_j')$ satisfying Eq. 3- 7. And the alternative intersection vector is denoted as $\mathbf{n}' = \mathbf{z}_i \times \mathbf{z}_j'$.

**Step 1:** Prove the intersection $\mathbf{n}$ is unique.

Substituting $\mathbf{n}'$ into Eq. 3 and Eq. 4 and subtracting the derived equations with Eq. 3 and Eq. 4, respectively, we can derive that

$$\mathbf{x}_i \cdot (\mathbf{n} - \mathbf{n}') = 0,$$
$$\mathbf{x}_i \times (\mathbf{n} - \mathbf{n}') \cdot \mathbf{z}_i = 0. \tag{8}$$

Since vectors $\mathbf{x}_i$ and $(\mathbf{n} - \mathbf{n}')$ are on the same plane perpendicular to $\mathbf{z}_i$, there exist $\lambda \neq 0$ such that

$$\mathbf{x}_i \times (\mathbf{n} - \mathbf{n}') = \lambda \mathbf{z}_i. \tag{9}$$

Then we can derive that

$$\lambda \mathbf{z}_i \cdot \mathbf{z}_i = 0. \tag{10}$$

Since $\lambda \neq 0$ and $\mathbf{z}_i$ is a unit vector, Eq. 10 creates a contradiction. Therefore, such $\mathbf{n}'$ does not exist. The intersection vector $\mathbf{n}$ of the planes $\mathbf{x}_i\mathbf{y}_i$ and $\mathbf{x}_j\mathbf{y}_j$ is uniquely determined by the Euler angle $\tau_{ji}^1$.

**Step 2:** Prove $\mathbf{z}_j$ is unique.

Substituting $\mathbf{z}'_j$ into Eq. 5 and subtracting the derived equation with Eq. 5, we can derive that

$$\mathbf{z}_i \cdot (\mathbf{z}_j - \mathbf{z}'_j) = 0. \tag{11}$$

Besides, based on the proof in **Step 1**, we have

$$\begin{aligned}
\mathbf{n} &= \mathbf{z}_i \times \mathbf{z}_j, \\
\mathbf{n} &= \mathbf{z}_i \times \mathbf{z}'_j.
\end{aligned} \tag{12}$$

By subtracting the above equations on both sides, we have

$$\mathbf{z}_i \times (\mathbf{z}_j - \mathbf{z}'_j) = 0. \tag{13}$$

Eq. 11 and Eq. 13 are contradicted since the non-zero vector $(\mathbf{z}_j - \mathbf{z}'_j)$ is both parallel and perpendicular to the unit vector $\mathbf{z}_i$. Thus, $z_j$ is uniquely determined by $\tau^1_{ji}$ and $\tau^2_{ji}$.

**Step 3:** Prove $\mathbf{x}_j$ is unique.

Substituting $\mathbf{x}'_j$ into Eq. 6 and Eq. 7 and subtracting the derived equations with Eq. 6 and Eq. 7, respectively, we can derive that

$$\begin{aligned}
\mathbf{n} \cdot (\mathbf{x}_j - \mathbf{x}'_j) &= 0, \\
\mathbf{n} \times (\mathbf{x}_j - \mathbf{x}'_j) \cdot \mathbf{z}_j &= 0.
\end{aligned} \tag{14}$$

As $(\mathbf{x}_j - \mathbf{x}'_j)$ and $\mathbf{n}$ are on the same plane perpendicular to $\mathbf{z}_j$, $\mathbf{n} \times (\mathbf{x}_j - \mathbf{x}'_j) = \mu \mathbf{z}_j$ holds for some $\mu \neq 0$. Thus, we can derive that

$$\mu \mathbf{z}_j \cdot \mathbf{z}_j = 0, \tag{15}$$

which is contradicted to the fact that $\mu \neq 0$ and $\mathbf{z}_j$ is a unit vector. Therefore, $\mathbf{x}_j$ can not have alternative solutions, *i.e.*, $\mathbf{x}_j$ is uniquely determined by $\tau^1_{ji}, \tau^2_{ji}, \tau^3_{ji}$.

**Step 4:** Prove $\mathbf{y}_j$ is unique.

Since $\mathbf{z}_j$ and $\mathbf{x}_j$ are unique, $\mathbf{y}_j = \mathbf{z}_j \times \mathbf{x}_j$ is also uniquely determined by the Euler angles.

The geometric representation $\mathcal{F}(G)_{\text{bb}} = \mathcal{F}(G)_{\text{base}} \cup \{(\tau^1_{ji}, \tau^2_{ji}, \tau^3_{ji})\}_{i=1,\dots,n,\, j \in \mathcal{N}_i}$ on backbone level provides unique representation for different protein backbone structures. Thus, the backbone level representation $\mathcal{F}(G)_{\text{bb}}$ is complete. $\qquad\square$

With the complete representation, we can compute the unique rotation matrix corresponding to the three static Euler angles $\tau^1_{ji}, \tau^2_{ji}, \tau^3_{ji}$ as

$$M = M_1 M_2 M_3, \tag{16}$$

where

$$M_1 = \begin{bmatrix} \cos\tau^1_{ji} & -\sin\tau^1_{ji} & 0 \\ \sin\tau^1_{ji} & \cos\tau^1_{ji} & 0 \\ 0 & 0 & 1 \end{bmatrix}, M_2 = \begin{bmatrix} 1 & 0 & 0 \\ 0 & \cos\tau^2_{ji} & -\sin\tau^2_{ji} \\ 0 & \sin\tau^2_{ji} & \cos\tau^2_{ji} \end{bmatrix}, M_3 = \begin{bmatrix} \cos\tau^3_{ji} & -\sin\tau^3_{ji} & 0 \\ \sin\tau^3_{ji} & \cos\tau^3_{ji} & 0 \\ 0 & 0 & 1 \end{bmatrix}.$$

Thus, given the unit vectors $(\mathbf{x}_i, \mathbf{y}_i, \mathbf{z}_i)$ of node $i$, we can derive the backbone coordinate system of node $j$ as

$$\begin{bmatrix} \mathbf{x}_j \\ \mathbf{y}_j \\ \mathbf{z}_j \end{bmatrix} = M \begin{bmatrix} \mathbf{x}_i \\ \mathbf{y}_i \\ \mathbf{z}_i \end{bmatrix}. \tag{17}$$

### B.2 Proof of the Completeness for All-Atom Level Representation

*Proof.* To prove completeness at the all-atom level, based on Def. 1, we need to prove the positions of all atoms in each amino acid are uniquely determined with $\mathcal{F}(G)_{\text{all}}$. Since the geometric representation at the backbone level is complete, the backbone structure of a given protein can be uniquely determined by $\mathcal{F}(G)_{\text{bb}}$. Therefore, for each amino acid $i$, we only need to prove that all the atoms of the side chain are uniquely determined by four side chain torsion angles. Note that all bond lengths and bond angles in each amino acid are fully rigid, thus we only consider the unit vector between two atoms

in an amino acid. Here we provide rigorous proof for the amino acid cysteine. The proof can be generalized to other types of amino acids.

A cysteine has six atoms, including $N, C_\alpha, C, O, C_\beta$, and $S_\gamma$. Firstly, the positions of $N, C_\alpha, C$, and $O$ are determined at the backbone level. We can easily further determine the position of $C_\beta$ since the atoms $N, C_\alpha, C$, and $C_\beta$ are in a rigid group as shown in Table 2 of (Jumper et al., 2021). Therefore, we only need to prove that the position of atom $S_\gamma$ is uniquely determined. For an amino acid cysteine with node index $i$, we use $\mathbf{p}_i^1, \mathbf{p}_i^2$, and $\mathbf{p}_i^3$ to denote the unit vectors of $\mathbf{r}_i^{C_\alpha} - \mathbf{r}_i^C, \mathbf{r}_i^{C_\beta} - \mathbf{r}_i^{C_\alpha}$, and $\mathbf{r}_i^{S_\gamma} - \mathbf{r}_i^{C_\beta}$. The unit vectors of $\mathbf{p}_i^1 \times \mathbf{p}_i^2$ and $\mathbf{p}_i^2 \times \mathbf{p}_i^3$ are denoted as $\mathbf{a}_i$ and $\mathbf{b}_i$. Given the side chain torsion angle $\chi_i^1$, we have

$$\mathbf{a}_i \cdot \mathbf{b}_i = \cos \chi_i^1,$$
$$\mathbf{a}_i \times \mathbf{b}_i \cdot \mathbf{p}_i^2 = \sin \chi_i^1. \tag{18}$$

Assume the position of atom $S_\gamma$ is not uniquely determined by $\chi_i^1$, then there is an alternative position of $S_\gamma$ satisfying Eq. 18. The new unit vector from $C_\beta$ to $S_\gamma$ is denoted as $\mathbf{p}_i^{3'}$, and $\mathbf{b}_i' = \mathbf{p}_i^2 \times \mathbf{p}_i^{3'}$. Substituting $\mathbf{b}_i'$ into Eq. 18 and subtracting the derived equations with Eq. 18, we can derive that

$$\mathbf{a}_i \cdot (\mathbf{b}_i - \mathbf{b}_i') = 0,$$
$$\mathbf{a}_i \times (\mathbf{b}_i - \mathbf{b}_i') \cdot \mathbf{p}_i^2 = 0. \tag{19}$$

Since vectors $\mathbf{a}_i$ and $\mathbf{b}_i - \mathbf{b}_i'$ are perpendicular to $\mathbf{p}_i^2$, $\mathbf{a}_i \times (\mathbf{b}_i - \mathbf{b}_i') = \rho \mathbf{p}_i^2$ holds for some $\rho \neq 0$. Then we can derive that

$$\mathbf{a}_i \times (\mathbf{b}_i - \mathbf{b}_i') \cdot \mathbf{p}_i^2 = \rho \mathbf{p}_i^2 \cdot \mathbf{p}_i^2 = 0. \tag{20}$$

Since $\rho \neq 0$ and $\mathbf{p}_i^2$ is a unit vector, Eq. 20 creates a contradiction. Therefore, such $\mathbf{p}_i^{3'}$ does not exist, and the position of atom $S_\gamma$ is uniquely determined by $\chi_i^1$. $\qquad\square$

## C  PRONET

In this section, we provide details about the geometric representations and the model architecture.

### C.1  GEOMETRIC REPRESENTATIONS

The geometric representation at the amino acid level is $\mathcal{F}(G)_{\text{base}} = \{(d_{ji}, \theta_{ji}, \phi_{ji}, \tau_{ji})\}_{i=1,\ldots,n,\, j \in \mathcal{N}_i}$ as introduced in Sec. 3.1. For each edge $ji$, we need to compute four geometries based on the positions of nodes $i, j, i-1, i+1, j-1$ and $j+1$. We use $\mathbf{p}_i^1, \mathbf{p}_i^2, \mathbf{p}_{ij}, \mathbf{p}_j^1, \mathbf{p}_j^2$ to denote the unit vectors of $\mathbf{r}_{i-1} - \mathbf{r}_i, \mathbf{r}_{i+1} - \mathbf{r}_i, \mathbf{r}_j - \mathbf{r}_i, \mathbf{r}_{j-1} - \mathbf{r}_j$ and $\mathbf{r}_{j+1} - \mathbf{r}_j$. Then the four geometries for edge $ji$ are computed based on

$$
\begin{aligned}
d_{ji} &= \|\mathbf{p}_{ij}\|_2, \\
\theta_{ji} &= \arccos\left(\mathbf{p}_i^1 \cdot \mathbf{p}_{ij}\right), \\
\mathbf{n}_1 &= \mathbf{p}_i^1 \times \mathbf{p}_i^2, \quad \mathbf{n}_2 = \mathbf{p}_i^1 \times \mathbf{p}_{ij}, \\
\phi_{ji} &= \operatorname{atan2}\left(\mathbf{n}_1 \cdot \mathbf{n}_2, \mathbf{n}_1 \times \mathbf{n}_2\right), \\
\mathbf{p}_i &= \begin{cases} \mathbf{p}_i^2, & \text{if } j = i-1 \\ \mathbf{p}_i^1, & \text{otherwise} \end{cases}, \quad
\mathbf{p}_j = \begin{cases} \mathbf{p}_j^2, & \text{if } i = j-1 \\ \mathbf{p}_j^1, & \text{otherwise} \end{cases}, \\
\mathbf{n}_3 &= \mathbf{p}_{ij} \times \mathbf{p}_i, \quad \mathbf{n}_4 = \mathbf{p}_{ij} \times \mathbf{p}_j, \\
\tau_{ji} &= \operatorname{atan2}\left(\mathbf{n}_3 \cdot \mathbf{n}_4, \mathbf{n}_3 \times \mathbf{n}_4\right).
\end{aligned}
\tag{21}
$$

As introduced in Sec. 3.2, the geometric representation at the backbone level is

$$
\begin{aligned}
\mathcal{F}(G)_{\text{bb}} &= \mathcal{F}(G)_{\text{base}} \cup \{(\tau_{ji}^1, \tau_{ji}^2, \tau_{ji}^3)\}_{i=1,\ldots,n,\, j \in \mathcal{N}_i} \\
&= \{(d_{ji}, \theta_{ji}, \phi_{ji}, \tau_{ji})\}_{i=1,\ldots,n,\, j \in \mathcal{N}_i} \cup \{(\tau_{ji}^1, \tau_{ji}^2, \tau_{ji}^3)\}_{i=1,\ldots,n,\, j \in \mathcal{N}_i} \\
&= \{(d_{ji}, \theta_{ji}, \phi_{ji}, \tau_{ji}, \tau_{ji}^1, \tau_{ji}^2, \tau_{ji}^3)\}_{i=1,\ldots,n,\, j \in \mathcal{N}_i}.
\end{aligned}
\tag{22}
$$

The steps to compute the Euler angles $\tau^1, \tau^2, \tau^3$ are provided in Sec. 3.2 and Fig. 2.

As introduced in Sec. 3.3, the geometric representation at the all-atom level is

$$\begin{aligned}\mathcal{F}(G)_{\text{all}} &= \mathcal{F}(G)_{\text{bb}} \cup \{(\chi_i^1, \chi_i^2, \chi_i^3, \chi_i^4)\}_{i=1,...,n} \\ &= \{(d_{ji}, \theta_{ji}, \phi_{ji}, \tau_{ji}, \tau_{ji}^1, \tau_{ji}^2, \tau_{ji}^3)\}_{i=1,...,n, \, j \in \mathcal{N}_i} \cup \{(\chi_i^1, \chi_i^2, \chi_i^3, \chi_i^4)\}_{i=1,...,n}.\end{aligned} \tag{23}$$

The atoms used to compute the side chain torsion angles $\chi^1, \chi^2, \chi^3, \chi^4$ are provided in Table 7.

## C.2 Model Architecture

**Overall architecture.** As shown in Fig. 4, the architecture of ProNet contains several interaction layers and an output layer. Each of the interaction layers updates node features based on the message passing scheme in Eq. 1. Specifically, for an interaction layer, the inputs are node features, edge features, and geometric representations, and the outputs are updated node features. Given the inputs, we firstly construct two intermediate updated node features. The first one is obtained by the Conv layer, and the second one is obtained by the Hier-Geom-MP layer. Then we concatenate these intermediate updated node features and use several fully-connected layers to obtain the final output of this interaction layer. Following interaction blocks, the final protein representation $\mathbf{g}$ is obtained with the output layer, which is implemented with a READOUT function:

$$\mathbf{g} = \text{READOUT}\left(\{\mathbf{v}_i^L\}_{i=1,...,n}\right). \tag{24}$$

Here, $\mathbf{v}_i^L$ indicates the feature vector of node $i$ at the last layer. Specifically, the Readout function includes a summation function and several fully-connected layers.

**Basis function.** We use basis functions to embed our proposed geometric representations. Specifically, we use spherical harmonics to encode distance $d$ and angles $\theta, \phi, \tau, \tau^1, \tau^2, \tau^3$ following Liu et al. (2022). Formally, $(d, \theta, \phi)$ is encoded with $j_\ell\left(\frac{\beta_{\ell n}}{c}d\right)Y_\ell^m(\theta, \phi)$, and $(d, \tau), (d, \tau^1), (d, \tau^2), (d, \tau^3)$ are encoded with $j_\ell\left(\frac{\beta_{\ell n}}{c}d\right)Y_\ell^0(a)$. Here $a$ can be $\tau, \tau^1, \tau^2$, or $\tau^3$. $j_\ell(\cdot)$ is a spherical Bessel function of order $\ell$, $Y_\ell^m$ is a spherical harmonic function of degree $m$ and order $\ell$, $c$ is the cutoff, and $\beta_{\ell n}$ is the $n$-th root of the Bessel function of order $\ell$. In addition, we use $\sin$ and $\cos$ functions to embed the side chain torsion angles $\chi^1, \chi^2, \chi^3, \chi^4$.

**Conv block.** The Conv block is adapted from the GraphConv layer (Morris et al., 2019) to update node features. Specifically, given input node features $\{\mathbf{v}_i^l\}_{i=1,...,n}$ and edge attributes $\{\mathbf{f}_{ij}\}_{i=1,...,n,j \in \mathcal{N}_i}$, the updating function is $\mathbf{m}_i^l = W_1\mathbf{v}_i^l + W_2\sum_{j \in \mathcal{N}_i}\mathbf{v}_j^l \odot (W_3\mathbf{f}_{ij})$. Here $\odot$ denotes element-wise multiplication, $\mathbf{f}_{ij}$ can be edge features $\mathbf{e}_{ij}$ or encoded geometric representations.

## D Dataset Description

**Fold Dataset.** We use the same dataset as in Hou et al. (2018) and Hermosilla et al. (2021). In total, this dataset contains 16,292 proteins from 1,195 folds. There are three test sets used to evaluate the generalization ability, namely Fold in which proteins from the same superfamily are unseen during training, Superfamily in which proteins from the same family are unseen during training, and Family in which proteins from the same family are present during training. Among the three test sets, Fold is the most difficult one since this test set differs the most from the training set. In this task, 12,312 proteins are used for training, 736 for validation, 718 for Fold, 1,254 for Superfamily, and 1,272 for Family.

**Reaction Dataset.** For reaction classification, the 3D structure for 37,428 proteins representing 384 EC numbers are collected from PDB (Berman et al., 2000), and EC annotations for each protein are downloaded from the SIFTS database (Dana et al., 2019). The dataset is split into 29,215 proteins for training, 2,562 for validation, and 5,651 for testing. Every EC number is represented in all 3 splits, and protein chains with more than 50% similarity are grouped together.

**LBA Dataset.** Following Somnath et al. (2021) and Townshend et al. (2021), we perform ligand binding affinity predictions on a subset of the commonly-used PDBbind refined set (Wang et al., 2004; Liu et al., 2015). The curated dataset of 3,507 complexes is split into train/val/test splits based on a 30% or 60% sequence identity threshold to verify the model generalization ability for unseen proteins. For a protein-ligand complex, we predict the negative log-transformed binding affinity $pK = -log_{10}(K)$ in Molar units.

Table 8: Model and training hyperparameters for our method on different tasks.

| Hyperparameter | Values/Search Space | | | |
|---|---|---|---|---|
| | Fold | Reaction | LBA | PPI |
| Number of layers | 3, 4, 5 | 3, 4, 5 | 3, 4, 5 | 3, 4, 5 |
| Hidden dim | 64, 128, 256 | 64, 128, 256 | 64, 128, 256 | 64, 128, 256 |
| Cutoff | 6, 8, 10 | 6, 8, 10 | 6, 8, 10 | 30 |
| Dropout | 0.2, 0.3, 0.5 | 0.2, 0.3, 0.5 | 0.2, 0.3 | 0 |
| Epochs | 1000 | 400 | 300 | 20 |
| Batch size | 16, 32 | 16, 32 | 8, 16, 32, 64 | 8, 16, 32 |
| Learning rate | 1e-4, 2e-4, 5e-4 | 1e-4, 2e-4, 5e-4 | 1e-5, 5e-5, 1e-4, 5e-4 | 1e-4, 2e-4, 5e-4 |
| Learning rate decay factor | 0.5 | 0.5 | 0.5 | 0.5 |
| Learning rate decay epochs | 100, 150, 200 | 50, 60, 70, 80 | 50, 70, 100 | 4, 8, 10 |

**PPI Dataset.** Following the dataset and experiment settings in Townshend et al. (2021), we predict whether two amino acids contact when the two proteins bind. We use the Database of Interacting Protein Structures (DIPS) (Townshend et al., 2019) for training and make prediction on the Docking Benchmark 5 (DB5) (Vreven et al., 2015). The split of protein complexes ensures that no protein in the training dataset has more than 30% sequence identity with any protein in the DIPS test set or the DB5 dataset.

## E  EXPERIMENTAL SETUP

This section describes the full experiment setup for each task considered in this paper. The implementation of our methods is based on the PyTorch (Paszke et al., 2019) and Pytorch Geometric (Fey & Lenssen, 2019), and all models are trained with the Adam optimizer (Kingma & Ba, 2014). All experiments are conducted on a single NVIDIA GeForce RTX 2080 Ti 11GB GPU. The search space for model and training hyperparameters are listed in Table 8. Note that we select hyperparameters at the amino acid, backbone, and all-atom levels by the same search space, and optimal hyperparameters are chosen by the performance on the validation set.

**Fold and Reaction dataset.** Similar to Hermosilla et al. (2021), we apply data augmentation techniques to increase data on fold and reaction classification tasks. Specifically, for the input data, we apply Gaussian noise with a standard deviation of 0.1 and anisotropic scaling in the range $[0.9, 1.1]$ for amino acid coordinates. The same noise is added to the atomic coordinates within the same amino acid, ensuring that the internal structure of each amino acid is not changed. We also mask the amino acid type with a probability of 0.1 or 0.2. For each interaction layer, we employ a Gaussian noise with a standard deviation of 0.025 to both features and Euler angles to further enhance the robustness of our models. We also find that warmup can further improve the performance on reaction classification.

**LBA dataset.** We follow the experiment settings in Somnath et al. (2021) for LBA tasks. Since our proposed methods focus on protein representation learning, we employ a two-branch network for a fair comparison. One branch of the network provides the representations for protein structures using our methods and the other branch generates the representations for ligands with a graph convolutional network. We employ the same architecture for the ligand branch as in Somnath et al. (2021) and use our models as the protein branch. A few fully-connected layers are then applied to the concatenations of protein and ligand representations to obtain the final representation of the corresponding complex.

## F  ADDITIONAL EXPERIMENTAL RESULTS

### F.1  RESULTS ON LBA

Results on LBA Dataset with standard deviation are listed in Table 9 and Table 10.

### F.2  RESULTS ON ADDITIONAL DATASETS FROM ATOM3D

We also conduct experiments on additional datasets from Atom3D (Townshend et al., 2021), specifically on Protein Structure Ranking (PSR) and Ligand Efficacy Prediction (LEP) datasets. Detailed descriptions of these datasets are provided below.

Table 9: Results with standard deviation on LBA dataset split by sequence identity 30%. The top two results are highlighted as **1st** and 2nd. * denotes methods trained with binding pockets only.

| Method | Sequence Identity 30% | | |
| --- | --- | --- | --- |
| | RMSE | Pearson | Spearman |
| Atom3D-3DCNN* (Townshend et al., 2021) | **1.416 ± 0.021** | 0.550 ± 0.021 | **0.553 ±0.009** |
| Atom3D-ENN* (Townshend et al., 2021) | 1.568 ± 0.012 | 0.389 ± 0.024 | 0.408 ± 0.021 |
| Atom3D-GNN* (Townshend et al., 2021) | 1.601 ± 0.048 | 0.545 ± 0.027 | 0.533 ± 0.033 |
| DeepDTA (Öztürk et al., 2018) | 1.866 ± 0.080 | 0.472 ± 0.022 | 0.471 ± 0.024 |
| Bepler and Berger (2019) (Bepler & Berger, 2019) | 1.985 ± 0.006 | 0.165 ± 0.006 | 0.152 ± 0.024 |
| TAPE (Rao et al., 2019) | 1.890 ± 0.035 | 0.338 ± 0.044 | 0.286 ± 0.124 |
| ProtTrans (Elnaggar et al., 2021) | 1.544 ± 0.015 | 0.438 ± 0.053 | 0.434 ± 0.058 |
| MaSIF (Gainza et al., 2020) | 1.484 ± 0.018 | 0.467 ± 0.020 | 0.455 ± 0.014 |
| GVP* (Jing et al., 2021a) | 1.594 ± 0.073 | - | - |
| IEConv (Hermosilla et al., 2021) | 1.554 ± 0.016 | 0.414 ± 0.053 | 0.428 ± 0.032 |
| Holoprot-Full Surface (Somnath et al., 2021) | 1.464 ± 0.006 | 0.509 ± 0.002 | 0.500 ± 0.005 |
| Holoprot-Superpixel (Somnath et al., 2021) | 1.491 ± 0.004 | 0.491 ± 0.014 | 0.482 ± 0.032 |
| ProNet-Amino Acid | 1.455 ± 0.009 | 0.536 ± 0.012 | 0.526 ± 0.012 |
| ProNet-Backbone | 1.458 ± 0.003 | 0.546 ± 0.007 | 0.550 ± 0.008 |
| ProNet-All-Atom | 1.463 ± 0.001 | **0.551 ± 0.005** | 0.551 ± 0.008 |

Table 10: Results with standard deviation on LBA dataset split by sequence identity 60%. The top two results are highlighted as **1st** and 2nd. * denotes methods trained with binding pockets only.

| Method | Sequence Identity 60% | | |
| --- | --- | --- | --- |
| | RMSE | Pearson | Spearman |
| Atom3D-3DCNN* (Townshend et al., 2021) | 1.621 ± 0.025 | 0.608 ± 0.020 | 0.615 ± 0.028 |
| Atom3D-ENN* (Townshend et al., 2021) | 1.620 ± 0.049 | 0.623 ± 0.015 | 0.633 ± 0.021 |
| Atom3D-GNN* (Townshend et al., 2021) | 1.408 ± 0.069 | 0.743 ± 0.022 | 0.743 ± 0.027 |
| DeepDTA (Öztürk et al., 2018) | 1.762 ± 0.261 | 0.666 ± 0.012 | 0.663 ± 0.015 |
| Bepler and Berger (2019) (Bepler & Berger, 2019) | 1.891 ± 0.004 | 0.249 ± 0.006 | 0.275 ± 0.008 |
| TAPE (Rao et al., 2019) | 1.633 ± 0.016 | 0.568 ± 0.033 | 0.571 ± 0.021 |
| ProtTrans (Elnaggar et al., 2021) | 1.641 ± 0.016 | 0.595 ± 0.014 | 0.588 ± 0.009 |
| MaSIF (Gainza et al., 2020) | 1.426 ± 0.017 | 0.709 ± 0.008 | 0.701 ± 0.001 |
| IEConv (Hermosilla et al., 2021) | 1.473 ± 0.024 | 0.667 ± 0.011 | 0.675 ± 0.019 |
| Holoprot-Full Surface (Somnath et al., 2021) | 1.365 ± 0.038 | 0.749 ± 0.014 | 0.742 ± 0.011 |
| Holoprot-Superpixel (Somnath et al., 2021) | 1.416 ± 0.022 | 0.724 ± 0.011 | 0.715 ± 0.006 |
| ProNet-Amino Acid | 1.397 ± 0.018 | 0.741 ± 0.008 | 0.734 ± 0.009 |
| ProNet-Backbone | 1.349 ± 0.019 | 0.764 ± 0.006 | 0.759 ± 0.001 |
| ProNet-All-Atom | **1.343 ± 0.025** | **0.765 ± 0.009** | **0.761 ± 0.003** |

**PSR (Protein Structure Ranking).** This task aims to predict the global distance test (GDT_TS) for each protein and is formulated as a regression task. In terms of the evaluation metrics, $R_S$ is Spearman correlation. Mean $R_S$ measures the correlation for structures corresponding to the same biopolymer, whereas global $R_S$ measures the correlation across all biopolymers. The results are listed in Table 11. As shown in the table, our methods can outperform all baseline methods and significantly improve mean $R_S$ results.

Table 11: Results on the PSR task. The top two results are highlighted as **1st** and 2nd.

| Method | Mean $R_S$ ↑ | Global $R_S$ ↑ |
| --- | --- | --- |
| Atom3D-3DCNN (Townshend et al., 2021) | 0.431 | 0.789 |
| Atom3D-GNN (Townshend et al., 2021) | 0.411 | 0.750 |
| GVP-GNN (Jing et al., 2021a) | 0.511 | 0.845 |
| ProNet-Amino Acid | 0.621 | 0.795 |
| ProNet-Backbone | **0.638** | 0.845 |
| ProNet-All-Atom | 0.632 | **0.849** |

**LEP (Ligand Efficacy Prediction).** This task aims to predict whether a molecule bound to the structures will be an activator of the protein's function or not. This task is formulated as a binary classification task, and the evaluation metric is AUROC. The results are listed in Table 12. As shown in the table, our methods can outperform all baseline methods.

Table 12: Results on the LEP task. The top two results are highlighted as **1st** and 2nd.

| Method | AUROC ↑ |
| --- | --- |
| Atom3D-3DCNN (Townshend et al., 2021) | 0.589 |
| Atom3D-GNN (Townshend et al., 2021) | 0.681 |
| Atom3D-ENN (Townshend et al., 2021) | 0.663 |
| GVP-GNN (Jing et al., 2021a) | 0.628 |
| ProNet-Amino Acid | 0.646 |
| ProNet-Backbone | 0.687 |
| ProNet-All-Atom | **0.692** |

