# OpenReview forum: "Learning Hierarchical Protein Representations via Complete 3D Graph Networks"
_ICLR.cc/2023/Conference — ICLR 2023 poster_

### Official Review · Reviewer_S9jc · 2022-10-21

**Confidence:** 3
**Correctness:** 3
**Technical Novelty And Significance:** 3
**Empirical Novelty And Significance:** 3
**Recommendation:** 8

**Clarity, Quality, Novelty And Reproducibility:**

The model architecture in Figure 4 is confusing. All three level representations are sent into the Hier-Geom-MP block. How to evaluate the contribution of each representation? In addition, $F(G)_{all}$ contains $F(G)_{bb}$ that contains $F(G)_{base}$. Why need to feed all of them to the Hier-Geom-MP block?

**Strength And Weaknesses:**

Strength: The hierarchical design of the representation improves the computational cost and model performance at the same time. Representations at different level can be used to solve different tasks.

Weaknesses: The all-atom level representation share $F(G)_{bb}$ with the backbone level representation. The backbone level representation share $F(G)_{base}$ with the amino acid level representation. The importance of such redundant design is unclear.

**Summary Of The Paper:**

This concerns learning the hierarchical representation (e.g., a.a., backbone, all-atom) for proteins with 3D structures. A novel hierarchical graph network, termed ProNet, was developed. ProNet is flexible, efficient and effective. It is shown that ProNet representations are complete at all levels if the base 3D graph network is complete. The experimental result shows the proposed model is able to achieve state-of-the-art for multiple tasks. It also shows that different downstream tasks my require different levels of representations.

**Summary Of The Review:**

This paper provides an effective way to represent proteins. Its idea is straightforward, and the results are promising. There are a few places that need more explanations.

---

> ### Author Response · Authors · 2022-11-13
> **Explained the motivation for our hierarchical design;  added details of model architecture (part 3)**
>
> > The model architecture in Figure 4 is confusing. All three level representations are sent into the Hier-Geom-MP block. How to evaluate the contribution of each representation?
>
> Sorry for the confusion. The Hier-Geom-MP block takes **either `one of the three levels`** of geometric information **as input** and generates the representation for **the corresponding level**.
>
> By feeding $\mathcal{F}(G)\_\text{base}$, $\mathcal{F}(G)\_\text{bb}$, **`or`** $\mathcal{F}(G)\_\text{all}$ into our network, we achieve three levels of ProNet, namely **ProNet-Amino Acid, ProNet-Backbone, and ProNet-All-Atom**.
>
> - Note that **ProNet-Amino Acid, ProNet-Backbone, ProNet-All-Atom** are the **`ablations` on different level geometric information** (`see results in Sec. 6`), and can be used to **evaluate the contribution of each geometric representation**.
>
>   Specifically, we have
>
>   - **ProNet-Amino Acid (Amino acid level)**: $\mathcal{F}(G)\_\text{base}=\{(d\_{ji}, \theta\_{ji}, \phi\_{ji}, \tau\_{ji})\}\_{i=1, \dots, n, \text{ } j\in \mathcal{N}\_i}$
>
>   - **ProNet-Backbone (Backbone level)**: $\mathcal{F}(G)\_\text{bb}=\mathcal{F}(G)\_{\text{base}} \cup \{(\tau^{1}\_{ji}, \tau^{2}\_{ji}, \tau^{3}\_{ji})\}\_{i=1, \dots, n, \text{ } j\in \mathcal{N}\_i}$
>
>   - **ProNet-All-Atom (All-atom level)**: $\mathcal{F}(G)\_\text{all}=\mathcal{F}(G)\_{\text{bb}} \cup \{(\chi^{1}\_i, \chi^{2}\_i, \chi^{3}\_i, \chi^{4}\_i)\}\_{i=1, \dots, n}$
>
>    **The comparison between ProNet-Amino Acid and ProNet-Backbone** shows the contribution of `Euler angles` $\{(\tau^{1}\_{ji}, \tau^{2}\_{ji}, \tau^{3}\_{ji})\}\_{i=1, \dots, n, \text{ } j\in \mathcal{N}\_i}$.
>
>    **The comparison between ProNet-Backbone and ProNet-All-Atom** shows the contribution of `side chain torsion angles` $\{(\chi^{1}\_i, \chi^{2}\_i, \chi^{3}\_i, \chi^{4}\_i)\}\_{i=1, \dots, n}$.
>
>    Based on the experimental results, we also observe that **different downstream tasks may require methods at different levels** (`Sec. 6.5`).
>
> **We have modified `Fig. 4` accordingly and added more details about network architecture in `Sec. 4` and `Appendix C`.**
>
> ---
>
> We sincerely thank you for your time! We look forward to your reply and further discussions, thanks!
>
> > References
>
>
> [1] Hermosilla, Pedro, et al. "Intrinsic-extrinsic convolution and pooling for learning on 3d protein structures." ICLR 2021.

---

> ### Author Response · Authors · 2022-11-13
> **Explained the motivation for our hierarchical design; added details of model architecture (part 2)**
>
> - **`Our hierarchical design contributes to high efficiency`**
>
>   Based on our hierarchical design, we always **treat each amino acid as a node**, even at the all-atom level. However, **existing all-atom methods treat each atom as a node**. The number of amino acids in a protein is much smaller than the number of atoms. Therefore, our method is more efficient. Specifically,
>
>     - **Ours: low complexity (`Table 1`)**: The complexity for previous all-atom level methods is `$O(Nk)$`, while that of our method is `$O(nk)$`. Here $n$, $N$, and $k$ denote the number of amino acids, the number of atoms, and the average degree in a 3D protein graph. And **$N\gg n$**.
>
>     - **Ours: fast training and inference (`Table 3`)**: Compared to the existing all-atom method IEConv [1], our method is about **5 times faster** (training and inference). In addition, all our experiments can be conducted on **a single NVIDIA GeForce RTX 2080 Ti 11GB GPU**.
>
>   With our efficient model, **researchers can quickly develop new models for protein-related tasks with limited computational resources**.
>
>
> - **`Our hierarchical design contributes to good performance`**
>
>   Benefitting from this hierarchical design, our methods **perform well on most datasets and evaluation metrics**.
>
>   - For the Fold dataset (`Sec. 6.1`), there are `three test sets`: `Fold`, `Sup.`, and `Fam.`, among which **`Fold` is the most difficult** one. This is because **this test set is the most different from the training data** (see detailed dataset description in `Appendix D`). Our ProNet achieves **significant improvement (accuracy from 48.3% to 52.7%)** on **the most difficult one**.
>
>     This demonstrates the **good generalization ability** of our method.
>
>
>     | Method        | React | Fold (`Fold`)  | Fold (`Sup.`) | Fold (`Fam.`) | Fold (Avg.) |
>     | ------------- | :-----: | :-----: | :----: | :----: | :---------: |
>     | SOTA (**all baseline methods**) | 87.2  | 48.3  | 70.3 | 99.5 | 72.7      |
>     | ProNet        | 86.4  | **52.7**  | **70.3** | 99.3 | **74.1**      |
>
>
>    - `LBA prediction task (Sec. 6.3)`: As summarized in the following table, ProNet **achieves SOTA on all 6 metrics (with improvements 1%, 8%, 10%, 2%, 2%, 3%) under `the same experimental setting`**. Note that we don't compare with Atom3D methods since **Atom3D uses different experiment settings and it is not fair to compare with them**. But we still include the results from Atom3D in our paper in case readers are interested in this setting. The models in **Atom3D** are trained with binding pocket structures, which makes the tasks **less challenging** (`see details in Sec. 6.3`).
>
>       | Method        | RMSE$\downarrow$ (30%) | Pearson$\uparrow$ (30%) | Spearman$\uparrow$ (30%) | RMSE$\downarrow$ (60%) | Pearson$\uparrow$ (60%) | Spearman$\uparrow$ (60%) |
>       | ------------- | :----------------------: | :-----------------------: | :------------------------: | :----------------------: | :-----------------------: | :------------------------: |
>       | SOTA (**all baseline methods w/o Atom3D**) | 1.464    | 0.509     | 0.500      | 1.365    | 0.749     | 0.742      |
>       | ProNet        | 1.455    | 0.551     | 0.551      | 1.343    | 0.765     | 0.761      |
>       | **Improve (%)**      | 1%       | 8%        | 10%        | 2%       | 2%        | 3%         |
>
>
>
>   - `PPI prediction task (Sec. 6.4)`: ProNet can **achieve SOTA** on this task.

---

> ### Author Response · Authors · 2022-11-14
> **Explained the motivation for our hierarchical design; added details of model architecture (part 1)**
>
> Dear Reviewer S9jc,
>
> Thanks for recognizing the high efficiency and good performance of our methods. And thanks for your valuable and constructive comments. We have revised the manuscript accordingly and also provide responses here.
>
> > The all-atom level representation share $\mathcal{F}(G)\_{bb}$ with the backbone-level representation. The backbone-level representation share $\mathcal{F}(G)\_{base}$ with the amino acid level representation. The importance of such redundant design is unclear.
>
> Thank you for your comment. We design such hierarchical levels based on `the hierarchical nature of protein structures`. The design is not redundant but necessary, as **in the hierarchical setting, learning at a higher level will need information from a lower level**. For example, in NLP, learning a representation of a sentence will need the representations of all the words that form this sentence. In addition, such design also results in `high efficiency` and `good performance` of our methods.
>
> - **`Hierarchical nature of protein structures`**
>
>   **We want to design the geometric representations** $\mathcal{F}(G)\_{base}$, $\mathcal{F}(G)\_{bb}$, and $\mathcal{F}(G)\_{all}$ **in `Sec. 3` following the hierarchical nature of protein structures** (as shown in **`Fig. 1`**).
>
>   In short, **as shown in the following table, we have** {$C_\alpha$} $\subset$ {$\{C_\alpha, N, C\}$} $\subset$ {$\{\text{All atoms}\}$}, therefore, **we design the geometric representations such that $\mathcal{F}(G)\_{base} \subset \mathcal{F}(G)\_{bb} \subset \mathcal{F}(G)\_{all}$**.
>
>   |    Levels             | Atoms we consider in each amino acid (`see details Fig. 1`) | Geometric representations (`Sec. 3`)|
>   | --------------------- | :-----:           | :-----:                    |
>   | Amino acid level      | $C_\alpha$        | $\mathcal{F}(G)\_{base}$   |
>   | Backbone level        | $C_\alpha, N, C$  | $\mathcal{F}(G)\_{bb}$     |
>   | All-atom level        | All atoms         | $\mathcal{F}(G)\_{all}$    |
>
>
>   Specifically, we use $C_\alpha$ coordinates, backbone atom ($C_\alpha, N, C$) coordinates, or all-atom coordinates to represent protein structures, leading to three levels of representations.
>
>   - At the **amino acid level**, we treat each amino acid as a node and use $C_\alpha$ coordinates as the position of the node.  This leads to the most coarse-grained representation of the protein. Then we design $\mathcal{F}(G)\_{base}$ to capture protein structures with $C_\alpha$ coordinates.
>
>   - At the **backbone level**, we consider $C_\alpha, N, C$ atoms for each amino acid to derive finer-grained protein representations. **Based on $\mathcal{F}(G)\_{base}$, we further use Euler angles** $\tau^{1}, \tau^{2}, \tau^{3}$ to capture $N, C$ atom information. Therefore, we design $\mathcal{F}(G)\_\text{bb}=\mathcal{F}(G)\_{\text{base}} \cup \{(\tau^{1}\_{ji}, \tau^{2}\_{ji}, \tau^{3}\_{ji})\}_{i=1, \dots, n, \text{ } j\in \mathcal{N}\_i}$.
>
>   - At the **all-atom level**, we consider all atoms in each amino acid. Given backbone structures, we only need to capture side chain structures. Therefore, **based on $\mathcal{F}(G)\_{bb}$, we further use side chain torsion angles** to capture side chain information. Thus we design $\mathcal{F}(G)\_\text{all}=\mathcal{F}(G)\_{\text{bb}} \cup \{(\chi^{1}\_i, \chi^{2}\_i, \chi^{3}\_i, \chi^{4}\_i)\}_{i=1, \dots, n}$.

---

### Official Review · Reviewer_hFzJ · 2022-10-23

**Confidence:** 3
**Correctness:** 4
**Technical Novelty And Significance:** 2
**Empirical Novelty And Significance:** 2
**Recommendation:** 6

**Clarity, Quality, Novelty And Reproducibility:**

The paper is well written and clear. The author did not provide code during reviewing period.

**Strength And Weaknesses:**

Strength:
This paper is well written. It also provide complete analysis of complete graph.
They conducted lots experiment to show the performance difference compared with previous methods.
They method also focus on amino acid rather than atoms that make both the data and the model to be light.

Weakness:
The method aims to integrate all level of protein information. But however, the performance increase is rather marginal compared with existing methods. That raised some question, whether such different level of information is needed. If so, how to integrate them. However, I do not think the author address these questions. For example, how about different integration methods.
This method has many components and lots of hyperparameters. I would recommend to conduct a more through ablation study. It could also bring some insights on why integrating different level of information.


**Summary Of The Paper:**

This paper proposed a framework, ProNet that integrated different level of protein information for protein related tasks. It showed improved performance on available datasets. ProNet is thus consisted by multiple components. The author also noticed that different tasks may require different level of representations.

**Summary Of The Review:**

I regard this paper is quite complete. But I find the contribution is quite marginal and lack some more in-depth analysis.

---

> ### Author Response · Authors · 2022-11-13
> **Explained the improvement of our method compared to previous SOTA methods; response to the questions; summarized our observations and insights (part 5)**
>
> We sincerely thank you for your time! We look forward to your reply and further discussions, thanks!
>
>
> > References
>
> [1] Gasteiger, Johannes, et al. "GemNet: Universal directional graph neural networks for molecules." NeurIPS 2021.
> [2] Hermosilla, Pedro, et al. "Intrinsic-extrinsic convolution and pooling for learning on 3d protein structures." ICLR 2021.
> [3] Hermosilla, Pedro, et al. "Contrastive representation learning for 3d protein structures."
> [4] Jing, Bowen, et al. "Learning from protein structure with geometric vector perceptrons." ICLR 2021.
> [5] Li, Jiahan, et al. "Directed Weight Neural Networks for Protein Structure Representation Learning."
> [6] Wang, Limei, et al. "ComENet: Towards Complete and Efficient Message Passing for 3D Molecular Graphs." NeurIPS 2022.
> [7] Zhang, Zuobai, et al. "Protein representation learning by geometric structure pretraining."

---

> ### Author Response · Authors · 2022-11-13
> **Explained the improvement of our method compared to previous SOTA methods; response to the questions; summarized our observations and insights (part 4)**
>
>   - Ablation between different levels also empirically show the importance of different levels
>
>     Note that our methods ProNet-Amino Acid, ProNet-Backbone, ProNet-All-Atom are the **`ablation` between different level information**.
>
>     In `Sec. 3`, we design Geometric representations at different levels as
>
>     - Amino acid level: $\mathcal{F}(G)\_\text{base}=\{(d_{ji}, \theta_{ji}, \phi_{ji}, \tau_{ji})\}_{i=1, \dots, n, \text{ } j\in \mathcal{N}_i}$
>
>     - Backbone level: $\mathcal{F}(G)\_\text{bb}=\mathcal{F}(G)\_{\text{base}} \cup \{(\tau^{1}\_{ji}, \tau^{2}\_{ji}, \tau^{3}\_{ji})\}_{i=1, \dots, n, \text{ } j\in \mathcal{N}_i}$
>
>     - All-atom level: $\mathcal{F}(G)\_\text{all}=\mathcal{F}(G)\_{\text{bb}} \cup \{(\chi^{1}\_i, \chi^{2}\_i, \chi^{3}\_i, \chi^{4}_i)\}\_{i=1, \dots, n}$
>
>     Then we feed these geometric features into our network ProNet (`Sec. 4`), resulting in **ProNet-Amino Acid, ProNet-Backbone, and ProNet-All-Atom**.
>
>       - **The comparison between ProNet-Amino Acid and ProNet-Backbone** shows the importance of `Euler angles` $\tau^{1}, \tau^{2}, \tau^{3}$.
>
>       - **The comparison between ProNet-Backbone and ProNet-All-Atom** shows the ability of `side chain torsion angles` $\chi^{1}, \chi^{2}, \chi^{3}, \chi^{4}$.
>
>       - We didn't do more ablations on $\mathcal{F}(G)\_\text{base}$ mainly because $d, \theta, \phi, \tau$ in $\mathcal{F}(G)\_\text{base}$ have been wildly used for small molecule representation learning [1,6] and shown to be important to capture structural information.
>
>   - How to integrate different level information (details about the network)?
>
>     As shown in `Fig. 4` and detailed in `Appendix C`, we use **basis functions** to embed the geometric representations and feed them into the network.
>     Specifically,
>
>       - $d, \theta, \phi, \tau$ and Euler angles $\tau^{1}, \tau^{2}, \tau^{3}$ are computed for each edges. And we use **spherical harmonics** (basis function) to encode them and use the embeddings as edge-level features.
>
>         **Spherical harmonics is a commonly used basis function to embed distances and angles, and is shown to be effective in previous methods** like GemNet [1] and ComENet [6].
>
>       - Since the side chain torsion angles $\chi^{1}, \chi^{2}, \chi^{3}, \chi^{4}$ are the features for each amino acid, and each amino acid is a node in our methods, **we add the embeddings of side chain torsion angles into the node attributes to incorporate the geometric information**.
>
>         Note that **adding the embeddings of angles into the node attributes is also used in some previous methods**. For example, GVP-GNN [4] and DWNN [5] add the $sin$ and $cos$ values of dihedral angles into node attributes to achieve backbone-level representation.
>
>
> > Summary of observations and insights
>
> - The most important insight of our paper is `it is necessary to incorporate hierarchical relations of proteins`.
>
>   - With the hierarchical design, our method is **much more accurate and efficient** than previous methods.
>
>   - Practically, high efficiency is also very important to the community. Proteins have much larger sizes compared with molecules. Also, in the current AI for Science era, the scale of real-world data is becoming larger and larger. **With our efficient model, researchers can quickly develop new models for protein-related tasks with limited computational resources**.
>
> - Secondly, previous methods only can achieve completeness **for small molecules**. However, Proteins are much larger and more complicated than molecules. We show that `it is possible and necessary to achieve completeness for proteins`.
>
> - In addition, an important observation from experimental results is `different downstream tasks may require methods at different levels` (`Sec. 6.5`). For researchers working on one specific task, our observation **provides guidelines for the design of their methods**. For example, for an interaction prediction task, they may need to pay more attention to all-atom level methods and design their method carefully to incorporate both backbone and side chain structures.

---

> ### Author Response · Authors · 2022-11-13
> **Explained the improvement of our method compared to previous SOTA methods; response to the questions; summarized our observations and insights (part 3)**
>
> > The method aims to integrate all level of protein information. But the performance increase is rather marginal compared with existing methods. That raised some question, whether such different level of information is needed. If so, how to integrate them.
>
> Thanks for your valuable comments and questions. These are also the main questions we want to answer in this paper.
>
>   - Why do we learn hierarchical representations of proteins (whether different level of information is needed)?
>
>     We design hierarchical levels based on `the inherently hierarchical nature of protein structures`. Besides, our hierarchical design also results in `high efficiency` and `good performance` of our methods. In addition, with such design, we can have `great flexibility` for different downstream tasks based on the `observations` (different downstream tasks may require methods at different levels).
>
>     - `Inherent hierarchical relations of proteins`:  We study protein representation learning based on the **unique structural properties of proteins**. Specifically, **there exist hierarchical relations among different levels** of protein representations (see **`Fig. 1 and the following table`**). Therefore, we design hierarchical protein representations to capture different level of information on protein structures.
>
>       |    Levels             | Atoms we consider in each amino acid (`see details in Fig. 1`) |
>       | --------------------- | :-----:           |
>       | Amino acid level      | $C_\alpha$        |
>       | Backbone level        | $C_\alpha, N, C$  |
>       | All-atom level        | All atoms         |
>
>     - `Good performance`: As detailed in our first reply, our hierarchical methods can achieve good performance compared with existing methods.
>
>     - `High efficiency`: our method can achieve great efficiency, especially at the **all-atom level**. Based on our hierarchical design, we always **treat each amino acid as a node**, even at the all-atom level. However, **existing all-atom methods treat each atom as a node**. The number of amino acids in a protein is much smaller than the number of atoms. Therefore, our method is more efficient. Specifically,
>
>       - `Low complexity`:  As shown in `Table 1`, **the complexity** for IEConv [2] and vector-gated GVP-GNN [4] (**all-atom methods**) is `$O(Nk)$`, while that of our method is `$O(nk)$`. Here $n$, $N$, and $k$ denote the number of amino acids, the number of atoms, and the average degree in a 3D protein graph. And **$N\gg n$**.
>
>       - `Fast training and inference`: as shown in `Table 3`, compared with the existing all-atom method IEConv [2], our method is about **5 times faster**. In addition, all our experiments can be conducted on **a single NVIDIA GeForce RTX 2080 Ti 11GB GPU**. With our efficient model, **researchers can quickly develop new models for protein related tasks with limited computational resources**.
>
>       - `Ablation results`: the ablation studies on all-atom level method in `Table 6` also show the efficiency (and effectiveness) of our all-atom level method.
>
>     - `The observation that different downstream tasks may require methods at different levels`: Importantly, from the experimental results, we observe that different downstream tasks may require methods at different levels. As discussed in `Sec. 6.5`, **ProNet-Backbone** can achieve good performance on function prediction tasks (`Sec. 6.1` and `Sec. 6.2`), while **ProNet-All-Atom** performs best on interaction prediction tasks (`Sec. 6.3` and `Sec. 6.4`). Our observation and different level methods can **provide guidelines for researchers** to design their methods for a specific task.
>
>     - `Great flexibility`: We focus on hierarchical representation learning of protein structures. However, **most existing methods ignore such hierarchical relations and only focus on one level**, which is not flexible and can hurt the performance and efficiency of existing methods. Our method can represent proteins at three levels, **possessing great flexibility for different data and downstream tasks**.
>
>     **In summary, our hierarchical design can incorporate `inherent hierarchical relations in protein structures` with `good performance`, `high efficiency`, and `great flexibility`. Therefore, it is necessary to learn hierarchical representations of proteins (at different levels).**

---

> ### Author Response · Authors · 2022-11-13
> **Explained the improvement of our method compared to previous SOTA methods; response to the questions; summarized our observations and insights (part 2)**
>
>   - Our method is **much more efficient** than current SOTA methods.
>
>     - As shown in `Table 3`, **SOTA methods on Fold (GearNet-Edge-IEConv [7])** and **React (IEconv [1] and new-IEConv [2])** suffer from **high computation complexity and memory cost**.
>
>       - **GearNet-Edge and GearNet-Edge-IEConv** require **a large amount of memory** and **cannot be trained using a single** Nvidia GeForce RTX 2080 Ti 11GB **GPU** (see `Sec 6.1 and Table 3`).
>
>       - **IEConv** has about **5 times more parameters** and is about **5 times slower** than our method (`Sec 6.2 and Table 3`).
>
>       - **New-IEConv**, similar to IEConv, is of high complexity and has about **9 times more parameters** (`Sec 6.2`).
>
>   In summary, our method can **achieve SOTA on most datasets**. However, **the second (or third) best performance on different datasets is shared by different baseline methods**. This indicates that previous baseline methods **do not generalize well**.  They usually achieve good performance on one dataset, but don't perform well on others. Besides, the improvement of our method over previous methods is **significant**, especially on some **important evaluation metrics**. In addition, our method is **much more efficient** than recent advanced methods. In short, our ProNet is both accurate and efficient due to our hierarchical and complete learning framework.

---

> ### Author Response · Authors · 2022-11-13
> **Explained the improvement of our method compared to previous SOTA methods; response to the questions; summarized our observations and insights (part 1)**
>
> Dear Reviewer hFzJ,
>
> Thanks for recognizing our contribution to complete (with proof) representations at three levels for protein representation learning. And thanks for your valuable and constructive comments. We have revised the manuscript accordingly and also provide responses here.
>
> > The improvement of the proposed method over previous baselines is relatively small.
>
>   Thank you for your comments.
>
>   - Firstly, we show that our method can achieve SOTA on **`almost all datasets`**. However, **the second (or third) best performance on different datasets is shared by different baseline methods**. This indicates that previous baseline methods **do not generalize well**. They usually achieve good performance on one dataset, but don't perform well on others.
>
>   - Secondly, our method can achieve **`significant improvements`** on some **important evaluation metrics**.
>
>   - Thirdly, our method is **`much more efficient`** than the current SOTA methods.
>
>   Here is the detailed analysis.
>
>   - Our method can achieve SOTA on **`almost all datasets`**. However, **the second (or third) best performance on different datasets is shared by different baseline methods**.
>
>     We listed **`the average rank`** of performance on all four tasks in the following table. \* denotes methods with high computation complexity (IEConv [1]).  Specifically,
>
>       - **Our methods are the best on** `Fold`, `LBA`, `PPI` tasks and `the average rank` over all tasks.
>
>       - In contrast, for previous methods,
>
>         - **IEConv** [1, 2] is the SOTA on `React` dataset but cannot achieve satisfying results on `Fold` and `LBA` tasks; Also, IEConv has high computation complexity.
>
>         - **GVP-GNN** [3, 4] is the second place for the `PPI` task but has poor performance on `React` and `Fold` tasks.
>
>
>     | Rank of performance | React | Fold  | LBA   | PPI   | Avg.   |
>     | :---- | :---: | :---: | :---: | :---: | :---: |
>     | **Ours-ProNet**     | 3     | **1** | **1** | **1** | **1.5** |
>     | IEConv* [1, 2]          | **1** | 3     | 4     | -    | 2.7     |
>     | HoloProt [6]         | 4     | -    | 2     | -    | 3     |
>     | GVP-GNN [3, 4]          | 7     | 7     | -    | 2     | 5.3     |
>
>   - Even compared with **previous SOTAs that were achieved by different baseline methods**, our ProNet can achieve **`significant improvements`**, especially on some **`important evaluation metrics`**.
>
>     - For the Fold dataset (`Sec. 6.1`), there are `three test sets`: `Fold`, `Sup.`, and `Fam.`, among which **`Fold` is the most difficult** one. This is because **this test set is the most different from the training data** (see detailed dataset description in `Appendix D`). Our ProNet achieves **significant improvement (accuracy from 48.3% to 52.7%)** on **the most difficult one**.
>
>       This demonstrates the **good generalization ability** of our method.
>
>
>       | Method        | Fold (`Fold`)  | Fold (`Sup.`) | Fold (`Fam.`) | Fold (Avg.) |
>       | ------------- |  :-----: | :----: | :----: | :---------: |
>       | Previous SOTA (**all baseline methods**) | 48.3%  | 70.3% | 99.5% | 72.7%      |
>       | ProNet        | **52.7%**  | **70.3%** | 99.3% | **74.1%**      |
>
>
>     - For LBA (`Sec. 6.3`), there are two splits. Our ProNet **achieves significant improvements (improve previous SOTA by 1%, 8%, 10%, 2%, 2%, 3%) on all splits (6 metrics)**.
>
>       In addition, `30% split is a harder evaluation setting than 60%`. This is because, for the 30% sequence identity split, **proteins in the test set** do not bear more than a 30 % **sequence similarity to proteins in the training set**. **The difference for proteins in the training and test sets is larger** than that on 60% split.
>
>       **ProNet improves more on the 30% splits (`more difficult`) than the 60% splits**. Specifically, on 30% split, the improvements are **1%, 8%, and 10%**, and on 60% split, the improvements are **2%, 2%, and 3%**.
>
>       | Method        | RMSE$\downarrow$ (30%) | Pearson$\uparrow$ (30%) | Spearman$\uparrow$ (30%) | RMSE$\downarrow$ (60%) | Pearson$\uparrow$ (60%) | Spearman$\uparrow$ (60%) |
>       | ------------- | :----------------------: | :-----------------------: | :------------------------: | :----------------------: | :-----------------------: | :------------------------: |
>       | Previous SOTA (**all baseline methods, w/o Atom3D `[*]`**) | 1.464    | 0.509     | 0.500      | 1.365    | 0.749     | 0.742      |
>       | ProNet        | 1.455    | 0.551     | 0.551      | 1.343    | 0.765     | 0.761      |
>       | **Improve (%)**      | 1%       | 8%        | 10%        | 2%       | 2%        | 3%         |
>
>           [*]: We don't include Atom3D methods since it is unfair to compare our results with Atom3D. The models in Atom3D are trained with binding pocket structures, which makes the tasks less challenging (see detailed explanation in Sec. 6.3).

---

### Official Review · Reviewer_X8gX · 2022-10-24

**Confidence:** 4
**Clarity, Quality, Novelty And Reproducibility:** The method description is relative cl…
**Correctness:** 4
**Technical Novelty And Significance:** 2
**Empirical Novelty And Significance:** Not applicable
**Recommendation:** 6

**Strength And Weaknesses:**

Strength:
1. The evaluation is fairly comprehensive. It included most of the state-of-the-art methods published recently.
2. Authors provide completeness proof for each layers, which is important for understanding the power of these geometric neural networks.

Weakness:
1. The improvement of the proposed method over previous baselines are relatively small. In particular, on the ligand binding affinity prediction task, the model underperforms the 3DCNN model (which is not equivariant) under 30% sequence identity cutoff. The 30% identity cutoff is a harder setup that requires more generalization to novel protein structures and therefore more important than 60% identity cutoff.

**Summary Of The Paper:**

This paper proposes a hierarchical graph neural network for protein property prediction. It models a protein at three levels -- amino acids, its backbone atoms, and its side chain atoms. Each level is modeled by a ComENet layer that learns a complete representation of the geometric structure. The method is evaluated on multiple protein classification tasks: fold classification, reaction classification, and ligand binding affinity prediction and compared with a variety of existing methods.

**Summary Of The Review:**

Overall, I vote for a weak acceptance of the paper because it provides a completeness proof for the proposed architecture, which can be valuable to the community.

---

> ### Author Response · Authors · 2022-11-13
> **Explained the improvement of our method compared to previous SOTA methods; clarified the experimental setting and result comparison on LBA. (Part 3)**
>
> We sincerely thank you for your time! We look forward to your reply and further discussions, thanks!
>
>
>
> > References
>
> [1] Hermosilla, Pedro, et al. "Intrinsic-extrinsic convolution and pooling for learning on 3d protein structures." ICLR 2021.
> [2] Hermosilla, Pedro, et al. "Contrastive representation learning for 3d protein structures."
> [3] Jing, Bowen, et al. "Learning from protein structure with geometric vector perceptrons." ICLR 2021.
> [4] Jing, Bowen, et al. "Equivariant graph neural networks for 3d macromolecular structure."
> [5] Panjkovich, Alejandro, et al. "Assessing the structural conservation of protein pockets to study functional and allosteric sites: implications for drug discovery." BMC structural biology 10.1 (2010): 1-14.
> [6] Somnath, Vignesh Ram, et al. "Multi-scale representation learning on proteins." NeurIPS 2021.
> [7] Zhang, Zuobai, et al. "Protein representation learning by geometric structure pretraining."

---

> > ### Comment · Reviewer_X8gX · 2022-11-17
> > **Thank you for your response**
> >
> > Thank you for your detailed response and revised manuscript. I think the results are more convincing than before so I will keep weak acceptance and raise my confidence score to 4.

---

> ### Author Response · Authors · 2022-11-13
> **Explained the improvement of our method compared to previous SOTA methods; clarified the experimental setting and result comparison on LBA. (Part 2)**
>
>   - Our method is **much more efficient** than current SOTA methods.
>
>     - As shown in `Table 3`, **SOTA methods on Fold (GearNet-Edge-IEConv [7])** and **React (IEconv [1] and new-IEConv [2])** suffer from **high computation complexity and memory cost**.
>
>       - **GearNet-Edge and GearNet-Edge-IEConv** require **a large amount of memory** and **cannot be trained using a single** Nvidia GeForce RTX 2080 Ti 11GB **GPU** (see `Sec 6.1 and Table 3`).
>
>       - **IEConv** has about **5 times more parameters** and is about **5 times slower** than our method (`Sec 6.2 and Table 3`).
>
>       - **New-IEConv**, similar to IEConv, is of high complexity and has about **9 times more parameters** (`Sec 6.2`).
>
>   In summary, our method can **achieve SOTA on most datasets**. However, **the second (or third) best performance on different datasets is shared by different baseline methods**. This indicates that previous baseline methods **do not generalize well**.  They usually achieve good performance on one dataset, but don't perform well on others. Besides, the improvement of our method over previous methods is **significant**, especially on some **important evaluation metrics**. In addition, our method is **much more efficient** than recent advanced methods. In short, our ProNet is both accurate and efficient due to our hierarchical and complete learning framework.
>
>
> > In particular, on the ligand binding affinity prediction task, the model underperforms the 3DCNN model (which is not equivariant) under the 30% sequence identity cutoff. The 30% identity cutoff is a harder setup that requires more generalization to novel protein structures.
>
> Thank you for your comments. Firstly, we show that the **experimental setting in Atom3D is different** from other methods (and our method), and **it is not fair** to compare our results with Atom3D. Then we analyze our results to show the **good generalization of our methods**.
>
> - `Different experimental settings in Atom3D`:
>
>   - Specifically, Atom3D methods only take the **binding pocket structure**, rather than the whole protein **as input**, making the task **less challenging**. This is because binding pockets are usually highly structural-conserved regions [5]. In their setting, the input structure varies less between the train and test datasets, providing additional information on the binding site, binding pose, and the interaction between protein and ligand. We refer to the original Atom3D paper for more details on input graph construction.
>
>   - Since our proposed methods focus on protein representation learning, **following HoloProt [6] (SOTA method on LBA)**, we employ a two-branch network for **a fair comparison** and encode the protein and ligand separately. Specifically, **we use the same ligand network as Holoprot and use our ProNet as the protein network**. In our setting, only 2D information is used on the ligand branch. We argue that our setting is more common in real-world applications like high-throughput screening. Our method is able to predict the binding affinity without considering the complex structure, which generally needs to be determined by X-ray crystallography or NMR spectroscopy and requires extensive experiments.
>
>   Therefore, **it is not fair to compare our results with Atom3D methods** (Atom3D-3DCNN, Atom3D-ENN, atom3D-GNN). But we still include the results from Atom3D in our paper in case readers are interested in this setting. Importantly, `We also list the reason in the second paragraph in Sec. 6.3` of our paper.
>
> - `Good results and generalization ability`:
>
>   - **Our method outperforms all baseline methods on both splits** with good improvements (**1%, 8%, 10%, 2%, 2%, and 3%** on 6 metrics).
>
>   - Specifically, on the 30% split, the improvements are **1%, 8%, and 10%**, and on the 60% split, the improvements are **2%, 2%, and 3%**. Comparing the improvements on these two splits, we can see **ProNet improves more on the 30% splits than the 60% splits**, which indicates the **good generalization ability** of our methods.

---

> ### Author Response · Authors · 2022-11-13
> **Explained the improvement of our method compared to previous SOTA methods; clarified the experimental setting and result comparison on LBA. (Part 1)**
>
> Dear Reviewer X8gX,
>
> Thanks for recognizing our contribution to complete (with proof) representations at three levels for protein representation learning. And thanks for your valuable and constructive comments. We have revised the manuscript accordingly and also provide responses here.
>
> > The improvement of the proposed method over previous baselines is relatively small.
>
>   Thank you for your comments.
>
>   - Firstly, we show that our method can achieve SOTA on **`almost all datasets`**. However, **the second (or third) best performance on different datasets is shared by different baseline methods**. This indicates that previous baseline methods **do not generalize well**. They usually achieve good performance on one dataset, but don't perform well on others.
>
>   - Secondly, our method can achieve **`significant improvements`** on some **important evaluation metrics**.
>
>   - Thirdly, our method is **`much more efficient`** than the current SOTA methods.
>
>   Here is the detailed analysis.
>
>   - Our method can achieve SOTA on **`almost all datasets`**. However, **the second (or third) best performance on different datasets is shared by different baseline methods**.
>
>     We listed **`the average rank`** of performance on all four tasks in the following table. \* denotes methods with high computation complexity (IEConv [1]).  Specifically,
>
>       - **Our methods are the best on** `Fold`, `LBA`, `PPI` tasks and `the average rank` over all tasks.
>
>       - In contrast, for previous methods,
>
>         - **IEConv** [1, 2] is the SOTA on `React` dataset but cannot achieve satisfying results on `Fold` and `LBA` tasks; Also, IEConv has high computation complexity.
>
>         - **GVP-GNN** [3, 4] is the second place for the `PPI` task but has poor performance on `React` and `Fold` tasks.
>
>
>         | Rank of performance | React | Fold  | LBA   | PPI   | Avg.   |
>         | :---- | :---: | :---: | :---: | :---: | :---: |
>         | **Ours-ProNet**     | 3     | **1** | **1** | **1** | **1.5** |
>         | IEConv* [1, 2]          | **1** | 3     | 4     | -    | 2.7     |
>         | HoloProt [6]         | 4     | -    | 2     | -    | 3     |
>         | GVP-GNN [3, 4]          | 7     | 7     | -    | 2     | 5.3     |
>
>   - Even compared with **previous SOTAs that were achieved by different baseline methods**, our ProNet can achieve **`significant improvements`**, especially on some **`important evaluation metrics`**.
>
>     - For the Fold dataset (`Sec. 6.1`), there are `three test sets`: `Fold`, `Sup.`, and `Fam.`, among which **`Fold` is the most difficult** one. This is because **this test set is the most different from the training data** (see detailed dataset description in `Appendix D`). Our ProNet achieves **significant improvement (accuracy from 48.3% to 52.7%)** on **the most difficult one**.
>
>       This demonstrates the **good generalization ability** of our method.
>
>
>       | Method        | Fold (`Fold`)  | Fold (`Sup.`) | Fold (`Fam.`) | Fold (Avg.) |
>       | ------------- |  :-----: | :----: | :----: | :---------: |
>       | Previous SOTA (**all baseline methods**) | 48.3%  | 70.3% | 99.5% | 72.7%      |
>       | ProNet        | **52.7%**  | **70.3%** | 99.3% | **74.1%**      |
>
>
>     - For LBA (`Sec. 6.3`), there are two splits. Our ProNet **achieves significant improvements (improve previous SOTA by 1%, 8%, 10%, 2%, 2%, 3%) on all splits (6 metrics)**.
>
>       In addition, `30% split is a harder evaluation setting than 60%`. This is because, for the 30% sequence identity split, **proteins in the test set** do not bear more than a 30 % **sequence similarity to proteins in the training set**. **The difference for proteins in the training and test sets is larger** than that on 60% split.
>
>       **ProNet improves more on the 30% splits (`more difficult`) than the 60% splits**. Specifically, on 30% split, the improvements are **1%, 8%, and 10%**, and on 60% split, the improvements are **2%, 2%, and 3%**.
>
>       | Method        | RMSE$\downarrow$ (30%) | Pearson$\uparrow$ (30%) | Spearman$\uparrow$ (30%) | RMSE$\downarrow$ (60%) | Pearson$\uparrow$ (60%) | Spearman$\uparrow$ (60%) |
>       | ------------- | :----------------------: | :-----------------------: | :------------------------: | :----------------------: | :-----------------------: | :------------------------: |
>       | Previous SOTA (**all baseline methods, w/o Atom3D `[*]`**) | 1.464    | 0.509     | 0.500      | 1.365    | 0.749     | 0.742      |
>       | ProNet        | 1.455    | 0.551     | 0.551      | 1.343    | 0.765     | 0.761      |
>       | **Improve (%)**      | 1%       | 8%        | 10%        | 2%       | 2%        | 3%         |
>
>           [*]: We don't include Atom3D methods since it is unfair to compare our results with Atom3D. The models in Atom3D are trained with binding pocket structures, which makes the tasks less challenging (see detailed explanation in Sec. 6.3).

---

### Official Review · Reviewer_KnXi · 2022-10-25

**Confidence:** 4
**Correctness:** 3
**Technical Novelty And Significance:** 2
**Empirical Novelty And Significance:** Not applicable
**Recommendation:** 3

**Clarity, Quality, Novelty And Reproducibility:**

Part 4 (and the ref to Appendix C) is very brief and lacks clarity, can be expanded to be more clear.
As I have mentioned above, the paper would benefit from more clarification of its novelty and several experiments.

**Strength And Weaknesses:**

This paper showcased a hierarchical graph network, "ProtNet", which computes the protein representations at different hierarchical levels. The authors investigated and integrated the physical protein (amino acid chains) conformational properties into their methods. Their experiments indicated the current method is largely on par with other SOTA methods, with ${maybe}$ marginal gain in some metrics.

1. The whole method is largely based on a very recent 2022 Neurips paper ComENet. Of course, ProtNet exploits unique properties of proteins and further adapted the ComENet to specialize for proteins. But the argument in 3.1 about "significantly different from ComENet" is not convincing enough and I would like to see the authors could further elaborate on the innovations and novelties in the current work.
 1.a) The first reason is ProNet is "based on the unique structural properties of protein". Thus, it is a special case of ComENet. Certainly, protein is a polymer of amino acids, thus providing more structural information for leveraging than general molecules (in which case we can also argue for greater generalization of the latter though). All properties (e.g., dihedral angles, side chain torsion) the authors discussed in the paper are long-known physical measures in protein characterization and have been exploited in various ML methods.
 1.b) Regarding ProtNet can represent backbones and all-atoms, this again has been seen in many previous ML methods for protein representations and while torsion angles can uniquely identify the conformation, I would like to understand better how simply adding them as node attributes archives "all-atom representation"
 1. c) The local coordinate system that is based on neighbor amino acids (they are linked by bonds) is a natural property of chains/sequences/polymers/proteins. The correct analogy should be atoms linked by chains in ComENet.
2. Moreover, I did not find any new discoveries, insights or learning opportunities in this work. The authors might be able to strength the paper by providing deeper analyses and understandings of the methods, and fundamentals ...
3. As the experiments largely followed (Townshend et al., 2021), it would be more convincing if we do all protein tasks exhaustively (e.g., RES, MSP, PSR, LEP are missing).
4. Additionally, in table 4, the 60% similarity cutoff is higher than the original 30% setup. (a) is there a reason we see higher RMSE for Atom3D? (b), doesn't it mean ProNet has less generalization power on more similar proteins?

**Summary Of The Paper:**

The authors proposed a complete protein learning method based on hierarchical representations: amino acids, backbone, and all-atom level.

**Summary Of The Review:**

The authors provide a new protein representation learning method. But I am mainly concerned about its novelty and significance.

---

> ### Author Response · Authors · 2022-11-13
> **Summarized novelty and insights; compared with related methods; clarified details of our method; conducted experiments on new datasets. (Part 6)**
>
> We sincerely thank you for your time! Hope the above responses address your questions and concerns. We look forward to your reply and further discussions, thanks!
>
> > References
>
> [1] Chaudhury, Sidhartha, et al. "PyRosetta: a script-based interface for implementing molecular modeling algorithms using Rosetta." Bioinformatics 2010.
> [2] Gasteiger, Johannes, et al. "GemNet: Universal directional graph neural networks for molecules." NeurIPS 2021.
> [3] Hermosilla, Pedro, et al. "Intrinsic-extrinsic convolution and pooling for learning on 3d protein structures." ICLR 2021.
> [4] Jing, Bowen, et al. "Learning from protein structure with geometric vector perceptrons." ICLR 2021.
> [5] Jing, Bowen, et al. "Equivariant graph neural networks for 3d macromolecular structure."
> [6] Jumper, John, et al. "Highly accurate protein structure prediction with AlphaFold." Nature 596.7873 (2021): 583-589.
> [7] Li, Jiahan, et al. "Directed Weight Neural Networks for Protein Structure Representation Learning."
> [8] Panjkovich, Alejandro, et al. "Assessing the structural conservation of protein pockets to study functional and allosteric sites: implications for drug discovery." BMC structural biology 10.1 (2010): 1-14.
> [9] Somnath, Vignesh Ram, et al. "Multi-scale representation learning on proteins." NeurIPS 2021.
> [10] Townshend, Raphael JL, et al. "Atom3D: Tasks on molecules in three dimensions." NeurIPS 2021.
> [11] Wang, Limei, et al. "ComENet: Towards Complete and Efficient Message Passing for 3D Molecular Graphs." NeurIPS 2022.
> [12] Zhang, Zuobai, et al. "Protein representation learning by geometric structure pretraining."

---

> > ### Author Response · Authors · 2022-11-17
> > **Reply to Reviewer KnXi**
> >
> > Dear Reviewer KnXi,
> >
> > We have summarized our novelty and insights, compared with related methods, clarified the details of our method, conducted experiments on new datasets from Atom3D, and revised the manuscript accordingly to address your concerns. The revised parts are highlighted in $\color{red}red$ in our revision.
> >
> > As it is approaching the end of the Author-Reviewer Discussion, could you take a look at the response and the revision, and let us know if we have addressed all your concerns? We are also happy to continue discussions if you still have remaining concerns.
> >
> > Thanks for your time!

---

> ### Author Response · Authors · 2022-11-13
> **Summarized novelty and insights; compared with related methods; clarified details of our method; conducted experiments on new datasets. (Part 5)**
>
> - About the results on LBA: in table 4, the 60% similarity cutoff is higher than the original 30% setup. (a) is there a reason we see higher RMSE for Atom3D? (b) doesn't it mean ProNet has less generalization power on more similar proteins? (revised `Sec. 6.3 and Table 4` in ${\color{red}red}$)
>
>   Thanks for your comments. Firstly, we want to emphasize that **the experimental setting in Atom3D is different** from other methods (and our method) in Table 4. Therefore, `it is not fair to compare with Atom3D`. But we still include the results from Atom3D in our paper in case readers are interested in this setting. Importantly, `We also list the reason in the second paragraph in Sec. 6.3` of our paper.
>
>   - Specifically, **Atom3D methods only take the `binding pocket structure`, rather than the whole protein `as input`**. Binding pockets are usually highly structural conserved regions [8]. In **their setting**, the input structure varies less between the train and test datasets, **providing additional information** on the binding site, binding pose, and the interaction between protein and ligand. We refer to the original Atom3D paper for more details on input graph construction.
>
>   - Since our proposed methods focus on protein representation learning, **following HoloProt [9] (SOTA method on LBA)**, we employ a two-branch network for **a fair comparison** and encode the protein and ligand separately. Specifically, **we use the same ligand network as Holoprot and use our ProNet as the protein network**. In our setting, only 2D information is used on the ligand branch. We argue that our setting is more common in **real-world applications** like high-throughput screening. Our method is able to predict the binding affinity without considering the complex structure, which generally needs to be determined by X-ray crystallography or NMR spectroscopy and requires extensive experiments.
>
>   **About 30% and 60% splits**:
>
>   - `30% protein sequence identity is a harder evaluation setting than 60%`. This is because, for the 30% sequence identity split, **proteins in the test set** do not bear more than a 30 % **sequence similarity to proteins in the training set**. The difference for proteins in the training and test sets is larger than that on the 60% split.
>
>   - In addition, `both splits are used in previous work`. Specifically, **for both splits, We use exactly the same dataset splits as in HoloProt [9] and Atom3D**.
>
>   **Is there a reason we see higher RMSE for Atom3D?**
>
>   - Our understanding of this question is: why the RMSE (the lower, the better) for Atom3D-3DCNN and Atom3D-ENN on the 60% split is higher than that on the 30% split? **If we misunderstand your question, please let us know**. Thanks.
>
>   - **This phenomenon in Atom3D is weird and it is different from all other methods in Table 4**. As we stated before, `30% protein sequence identity is a harder evaluation setting than 60%`. For other methods in Table 4, the results on the 60% split are always better than that on the 30% split. `Unfortunately, this problem is not discussed in the Atom3D paper`. *In addition, there is also a raised issue on the GitHub of Atom3D, but it is not answered by the authors*. Based on our understanding, `one possible explanation is` that Atom3D methods only take the binding pocket structure, rather than the whole protein as input. Compared with full proteins, the difference between the pockets in train/test on the 30% split and that on the 60% split is relatively smaller. Therefore, the generalization gap for the 30% split in the Atom3D setting is relatively smaller than that in our experimental setting. However, we study protein representation learning, so our model takes the whole protein as input, which is consistent with other baselines.
>
>   **Doesn't it mean ProNet has less generalization power on more similar proteins?**
>
>   - **Good performance of our method**: Since **it is not fair to compare with Atom3D methods**, in this part, **we only compare with other baseline methods** in Table 4. **Our method outperforms all baseline methods on both splits (6 evaluation metrics)**. The improvements on the 6 metrics are **1%, 8%, 10%, 2%, 2%, 3%**, respectively.
>
>   - Specifically, on the 30% split, the improvements are **1%, 8%, and 10%**, and on the 60% split, the improvements are **2%, 2%, and 3%**. Comparing the improvements on these two splits, we can see **ProNet improves more on the 30% splits (a harder task) than the 60% splits**, which indicates the **good generalization ability** of our methods.
>
>   Based on `our analysis of Atom3D` and the `good performance of our method` (improving previous SOTA results `on both 30% and 60% splits`), we respectfully disagree with the reviewer's comment on the generalization ability of ProNet.
>
>   `We have revised Sec. 6.3 to make this clearer to readers.`

---

> ### Author Response · Authors · 2022-11-13
> **Summarized novelty and insights; compared with related methods; clarified details of our method; conducted experiments on new datasets. (Part 4)**
>
> > Experiments: the datasets we used, the results on LBA
>
> - About the dataset we used, should consider all protein tasks in Atom3D [10]
>
>   Thanks for your comments. We first explain that **`the datasets we used in the paper are enough` to test the performance of protein representation learning methods**. Then we show **`the results on newly added datasets`**.
>
>   - We follow IEConv [3], HoloProt [9], and Atom3D [10] and conduct experiments on four datasets. Systematically, the tasks in the four datasets can be **`categorized into three classes`**, namely property prediction for **a single protein** (`Sec. 6.1, 6.2`), **protein-ligand interaction** (`Sec. 6.3`), and **protein-protein interaction** (`Sec. 6.4`). These three classes cover the most commonly used protein tasks. For example, PSR in Atom3D belongs to the first class; LEP in Atom3D belongs to the second class.
>
>     **Given the comprehensiveness of the tasks in our paper, we believe the datasets we used are enough to test the performance of protein representation learning methods.**
>
>   - However, we still think your concern is insightful and Atom3D is a good benchmark with newly collected data and designed protein tasks. Therefore, we **conducted experiments on additional datasets** in Atom3D. As you suggested, there are four additional protein tasks (RES, MSP, PSR, LEP) in Atom3D.
>
>     - PSR (Protein Structure Ranking): This task aims to predict the global distance test (GDT_TS) for each protein, therefore it can be **`categorized into` the property prediction for a single protein** class. This task is formulated as a regression task. In terms of the evaluation metrics, $R_S$ is Spearman correlation. Mean measures the correlation for structures corresponding to the same biopolymer, whereas global measures the correlation across all biopolymers. The results are listed in the following table. As shown in the table, **our methods can outperform all baseline methods and significantly improve `mean R_S` results**.
>
>        Method        | mean $R_S$ | glob. $E_S$  |
>       | ------------- | :-----: | :-----: |
>       | Atom3D-3DCNN [10]     | 0.431  | 0.789  |
>       | Atom3D-GNN [10]     | 0.411  | 0.750  |
>       | GVP-GNN [4]      | 0.511  | **0.845**  |
>       | ProNet-Amino Acid| **0.621**  | 0.795  |
>       | ProNet-Backbone  | **0.638**  | **0.845**  |
>       | ProNet-All-Atom  | **0.632**  | **0.849**  |
>
>
>     - LEP (Ligand Efficacy Prediction): This task aims to predict whether a molecule bound to the structures will be an activator of the protein’s function or not, therefore it can be **`categorized into` the protein-ligand interaction** class. This task is formulated as a binary classification task, and the evaluation metric is AUROC. The results are listed in the following table. As shown in the table, **our methods can outperform all baseline methods.**
>
>        Method        | mean $R_S$ |
>       | ------------- | :-----: |
>       | Atom3D-3DCNN [10]     | 0.589  |
>       | Atom3D-GNN [10]       | 0.681  |
>       | Atom3D-ENN [10]       | 0.663  |
>       | GVP-GNN [4]      | 0.628  |
>       | ProNet-Amino Acid| 0.646  |
>       | ProNet-Backbone  | **0.687**  |
>       | ProNet-All-Atom  | **0.692**  |
>
>     - RES (Residue Identification): In short, we focus on **`protein` representation learning**, but **`each data sample in this dataset is only a small part of a protein`**, therefore, **we didn't conduct experiments on this dataset**. Specifically, this task aims to predict the identity of particular masked amino acid based on its local surrounding structure. Each instance in this dataset represents **a fragment extracted from a protein structure**. In addition, this dataset is **`used for pre-training`** in a recent work [5].
>
>     - MSP (Mutation Stability Prediction): In short, we hope to test our performance on **`real-world proteins`**, but **`the data in this dataset are based on some simulation tools`**, therefore, **we didn't conduct experiments on this dataset**. Specifically, This task aims to predict whether a mutation stabilizes the protein complex. The mutagenesis in this dataset is obtained by **in silico simulation** with PyRosseta [1] (as explained in `Appendix C.1 in the Atom3D paper` [10]), thus might vary from the actual case in wet-lab experiments.
>
>   **We can put the results on new datasets in the Appendix of the paper if you suggest so.**

---

> ### Author Response · Authors · 2022-11-13
> **Summarized novelty and insights; compared with related methods; clarified details of our method; conducted experiments on new datasets. (Part 3)**
>
> > Details of our method: the local coordinate system in Sec. 3.1, the all-atom level method in Sec. 3.3, the model in Sec. 4 and Appendix C
>
> - Question: About the local coordinate system, the correct analogy should be atoms linked by chains in ComENet
>
>   Thanks for your comments. There's some misunderstanding here.
>
>   Firstly, we want to emphasize that **we treat `each amino acid as a node`, not an atom as a node** (the case in ComENet).
>
>   Since we treat each amino acid as a node, **`the nodes are linked to form a chain` (the important property of proteins)**. But for atoms at small molecules (**ComENet is designed for**), it is totally different. **`The atoms at small molecules are not linked to a chain`** (it is more like a regular graph where each atom is connected to several atoms).
>
>   Therefore, we build the local coordinate system for each amino acid **efficiently based on the chain property** of proteins. But for small molecules without such chain property, ComENet builds the local coordinate system **in a more complicated way (with extra computation)**.
>
>   Specifically,
>
>     - We treat each amino acid as a node and use Ca coordinates as the position of the node. Thus **we have nodes with indexes $1, 2, ...,n$ and positions $p\_1, p\_2, ..., p\_n$.**
>     - **In our method**, for a node $i$, we use $p\_{i-1}, p\_i, p\_{i+1}$ to define the local coordinate system.
>     - **If we directly use ComENet**: for a node $i$, they use $p\_{f\_i}, p\_i, p\_{s\_i}$ to define the local coordinate system. Here $f\_i$ and $s\_i$ are the nearest (with the shortest distance) node and second nearest node of node $i$, which requires **extra computation** to sort the distances and find the nearest two.
>
>   In summary, we **use a different way to define the local coordinate system for each node** (efficient and based on protein properties), **`not directly analogizing amino acids in our methods to atoms in ComENet`**.
>
> - Question: About our all-atom level method, while torsion angles can uniquely identify the conformation, I would like to understand better how simply adding them as node attributes archives all-atom representation.
>
>   - Firstly, we want to particularly point out that our method can achieve all-atom representation from the **`geometric perspective`**.
>
>     We `prove in Sec. 3.4 and Appendix B` that **the geometric representation $\mathcal{F}(G)\_\text{all}$ is `complete`**, which means $\mathcal{F}(G)\_\text{all}$ can capture the geometric information of all the atoms in a protein. This is exactly what we mean that: our method achieves all-atom level representations from the geometric perspective.
>
>     The reason why $\mathcal{F}(G)\_\text{all}$ can capture all-atom information is:
>
>      - At the amino acid level, we design a complete $\mathcal{F}(G)\_\text{base}$ that can capture amino acid level information.
>
>      - **Based on the complete $\mathcal{F}(G)\_\text{base}$**, we design $\mathcal{F}(G)\_\text{bb}=\mathcal{F}(G)\_\text{base} \cup \{(\tau^1_{ji}, \tau^2_{ji}, \tau^3_{ji})\}_{i=1, \dots, n, \text{ } j\in \mathcal{N}\_i}$ that can capture backbone level information (**completeness at the backbone level**, `Sec. 3.2 and 3.4`).
>
>      - **Based on the complete $\mathcal{F}(G)\_\text{bb}$**, at the all-atom level, the geometric representation is $\mathcal{F}(G)\_\text{all}=\mathcal{F}(G)\_{\text{bb}} \cup \{(\chi^{1}\_i, \chi^{2}\_i, \chi^{3}\_i, \chi^{4}\_i)\}\_{i=1, \dots, n}$. Here $\chi$ are side chain torsion angles.
>
>        Given $\mathcal{F}(G)\_{\text{bb}}$, we assume all bond lengths and angles in each amino acid are rigid following AlphaFold [6], then **`the only degree of freedom` we need to consider is torsion angles in side chains**.
>
>        Therefore, **by further considering side chain torsion angles based on $\mathcal{F}(G)\_\text{bb}$, $\mathcal{F}(G)\_\text{all}$ can capture all-atom information. Thus our method can achieve all-atom representation from `geometric perspective`.**
>
>   - Then we consider `how to incorporate the geometric representations into the network`.
>
>     - Since the **side chain torsion angles are the features for each amino acid**, and each amino acid is a node in our methods, **we add the embeddings of side chain torsion angles `into node attributes` to incorporate the geometric information**.
>
>     - **`Adding embeddings of angles into node attributes` is also used in many previous protein representation learning methods**. For example, GVP-GNN [4] and DWNN [7] add the $sin$ and $cos$ values of dihedral angles into node attributes to achieve backbone level representation.
>
> - Part 4 (and the ref to Appendix C) is very brief and lacks clarity, can be expanded to be more clear.
>
> **We have revised Sec. 4, Fig. 4, and Appendix C** in ${\color{red}red}$ to make this clear.
>   For Sec. 4, we mainly edited the figure the make the flow much clearer.
>   For Appendix C, we added more details of the model architecture.

---

> ### Author Response · Authors · 2022-11-13
> **Summarized novelty and insights; compared with related methods; clarified details of our method; conducted experiments on new datasets. (Part 2)**
>
> > Discussions with ComENet [11] (revised Sec. 3.1 in ${\color{red}red}$)
>
> - `Problem study`: We study **protein representation learning** considering the inherent hierarchical information, while ComENet is designed for **small molecules**. Structures of small molecules (for QM9, a commonly used data, the **average number of atoms is 18**) are `much less complicated` than protein structures (for React data in our paper, the **average number of amino acids is 300, and the largest number is 3615**).
>
> - Our main novelty is designing a **hierarchical protein learning** framework to incorporate protein structures at three levels. We then aim to achieve geometric completeness at all three different levels, and **ComeNet can `only be used to` achieve geometric representation `at the amino acid level`**.
>
> - Although the representation at the **amino acid level** is based on ComENet, **`the backbone and all-atom levels` are totally different from ComENet**. We carefully design these two levels based on the unique structural properties of proteins.
>
> - Even at the amino acid level, the used method is slightly different from ComENet. Technically, when computing $\mathcal{F}(G)\_\text{base}$, we use **a different strategy to define the `local coordinate system`** for each node $i$ (See `Sec. 3.1` and `our Part3 response`).
>
> - In addition to geometric representations, **our `model architecture` is also significantly different from ComENet**. we **design our novel model architecture in `Sec. 4`** to learn hierarchical protein representations. Particularly, our hierarchical message passing `Hier-Geom-MP` is **specially designed for hierarchical protein representation learning and can effectively capture hierarchical relations of proteins** as shown in `Fig. 4`. *For clarity, we have revised Sec. 4 and Appendix C in ${\color{red}red}$ with additional details on model architecture*.
>
> - Note that although we design $\mathcal{F}(G)\_\text{base}$ based on ComeNet, **ComENet is not the only solution** to achieve completeness at the amino acid level. For example, we can also use GemNet [2] to design $\mathcal{F}(G)\_\text{base}$. **Due to the high efficiency of ComENet**, we finally design our $\mathcal{F}(G)\_\text{base}$ based on ComeNet.
>
> > Discussions with other ML methods (revised Sec. 3.3 in ${\color{red}red}$)
>
> Firstly, we want to emphasize that we design a **hierarchical protein learning** framework, which can incorporate hierarchical protein structures completely and efficiently, while **previous methods for protein representation learning** either ignore hierarchical relations within proteins or suffer from excessive computational complexity, as summarized in `Table 1`.
>
> - About `dihedral angles` and `backbone level representations`
>
>   - **Backbone dihedral angles** are important properties of protein structures, and **it is used in several papers to achieve backbone level representations**. For example, GVP-GNN [4] includes the $sin$ and $cos$ values of dihedral angles as part of node features to represent protein backbones.
>   - `We use Euler angles, instead of using dihedral angles`, to achieve backbone level representation. In `Sec. 3.2`, we discuss **`the advantages of using Euler angles` compared to using dihedral angles**. In short, our method can significantly improve the learning efficiency **by computing much less angles (3 versus 3$n$ where $n$ is the number of amino acids).**
>   - Additionally, as summarized in `Table 1`, `existing backbone-level methods` only capture **one level of protein structures**, ignoring hierarchical relations within proteins. However, **our method can represent proteins at three levels** with great flexibility for different data sources and downstream tasks.
>
> - About `side chain torsion angles` and `all-atom level representations`
>
>   - Although side chain torsion angles are important properties of protein structures and are used in many ML methods (e.g. AlphaFold [6] uses side chain torsion angles to predict all-atom coordinates), to the best of our knowledge, **none of the `existing protein representation learning methods` use side chain torsion angles to capture the geometric information at the all-atom level**. We propose to use side chain torsion angles to **efficiently** achieve geometric completeness at the all-atom level.
>
>   - Based on our knowledge, **there are only two existing protein representation learning methods that can achieve all-atom representation**, as summarized in `Table 1`. And **they all suffer from `excessive computational complexity`** (`$O(Nk)$`) since they all treat each atom as a node. But **our all-atom method is much more efficient** (with complexity `$O(nk)$`) by treating each amino acid as a node. **This great efficiency is partially realized by integrating side chain torsion angles at the all-atom level.** Here $n$, $N$, and $k$ denote the number of amino acids, the number of atoms, and the average degree in a 3D protein graph, and $N\gg n$.

---

> ### Author Response · Authors · 2022-11-13
> **Summarized novelty and insights; compared with related methods; clarified details of our method; conducted experiments on new datasets. (Part 1)**
>
> Dear Reviewer KnXi, thanks for your comments!
>
> We have **conducted experiments on new datasets** from Atom3D and revised the manuscript accordingly. We also provide responses here.
>
> > Summary of our motivation and novelty
>
>   - We focus on **hierarchical representations learning of protein structures**. The **`main motivation`** is **there exist hierarchical relations among different levels** of protein representations (see `Fig. 1`, amino acid, backbone, and all-atom levels). However, **most existing methods ignore such hierarchical relations** and only focus on one level, which is not flexible and can hurt both the performance and efficiency. Due to the absence of the hierarchical relations, existing methods are not powerful. Without building a higher level on top of a lower level, existing methods are usually computationally expensive. We demonstrate (both theoretically and empirically) that integrating such hierarchical relations can boost performance, efficiency, and scalability.
>
>   - We answer three important questions in this paper: (1) How to integrate hierarchical information? (2) What geometric information should we use to fully capture protein structures at hierarchical different levels? (3) How to incorporate the information such that the whole pipeline is efficient?
>
> The **novelty and contributions** of our ProNet can be summarized into the following aspects:
>
>   - `Hierarchical protein representations`: Our method can incorporate **inherent hierarchical relations** in protein structures and largely advance protein representation learning (`good performance`). The representation at the higher level is built on top of the lower level such that we treat each amino acid as a node, while previous methods don't have a such property (`high efficiency`). In addition, different practical tasks require representations at different levels (`high flexibility`).
>
>   - `Complete geometric representations at three levels`: this enables models to generate informative and discriminative representations.
>
>   - `Model architecture`: Based on the geometric representations $\mathcal{F}(G)\_\text{base}$, $\mathcal{F}(G)\_\text{bb}$, $\mathcal{F}(G)\_\text{all}$, we **design our novel model architecture in `Sec. 4`** to learn hierarchical protein representations. Particularly, our hierarchical message passing `Hier-Geom-MP` is **specially designed for protein learning and can effectively capture hierarchical relations of proteins**.
>
> - `High efficiency`: Our method is **much more efficient** than recent advanced methods like IEConv [3] and GearNet-Edge [12].
>   - `Theoretically, low complexity`: As shown in `Table 1`, **the complexity** for IEConv [3] and vector-gated GVP-GNN [5] is `$O(Nk)$`, while that of our method is `$O(nk)$`. Here, $n$, $N$, and $k$ denote the number of amino acids, the number of atoms, and the average degree in a 3D protein graph, respectively. Note that **$N\gg n$**.
>     - `Practically, less memory, fast training and inference`: As shown in `Table 3`, current SOTA methods either **require a large amount of memory** (GearNet-Edge [12] and GearNet-Edge-IEConv [12], they can't train the model using one Nvidia GeForce RTX 2080 Ti 11GB GPU) or are about **5 times slower** (IEConv [3]) than our method. in addition, all our experiments can be conducted on **a single NVIDIA GeForce RTX 2080 Ti 11GB GPU**.
>
>     - `Efficiency of our all-atom level method`: The ablation studies on the all-atom level method in `Table 6` also show the efficiency (and effectiveness) of our all-atom level method.
>
>
>
> > Summary of discoveries and insights
>
> - The most important insight of our paper is `it is necessary to incorporate hierarchical relations of proteins`.
>
>   - With the hierarchical design, our method is **much more accurate and efficient** than previous methods.
>
>   - Practically, high efficiency is also very important to the community. Proteins have much larger sizes compared with molecules. Also, in the current AI for Science era, the scale of real-world data is becoming larger and larger. **With our efficient model, researchers can quickly develop new models for protein-related tasks with limited computational resources**.
>
> - Secondly, previous methods only can achieve completeness **for small molecules**. However, Proteins are much larger and more complicated than molecules. We show that `it is possible and necessary to achieve completeness for proteins`.
>
> - In addition, an important observation from experimental results is `different downstream tasks may require methods at different levels` (`Sec. 6.5`). For researchers working on one specific task, our observation **provides guidelines for the design of their methods**. For example, for an interaction prediction task, they may need to pay more attention to all-atom level methods and design their method carefully to incorporate both backbone and side chain structures.

---

> ### Author Response · Authors · 2022-11-20
> **Reviewer KnXi: could you check our response and revision?**
>
> Reviewer KnXi,
>
> We posted our detailed response a week ago. We also added more experiments and substantially revised the paper based on your comments. Could you read and let us know if there are more questions? Thanks.
>
> Authors

---

### Decision · Program_Chairs · 2023-01-20

**Decision:**

Accept: poster

**Justification For Why Not Higher Score:**

While the method is novel and experiments promising, the contributions are not groundbreaking as they build on prior architectures and protein features.

**Justification For Why Not Lower Score:**

The approach is very interesting to the community, will inspire researchers in the growing area of protein representation learning.  The authors convincingly demonstrate great performance and efficiency compared to existing approaches. The authors have satisfactorily addressed all concerns raised in the reviews.

**Metareview: Summary, Strengths And Weaknesses:**

The paper proposes an approach to perform hierarchical representation learning of protein structures, where are each level a complete structure representation is learnt. Extensive experiments are performed that demonstrate the performance and efficiency of the approach and illustrates how different tasks may benefit from different levels.

The paper makes some interesting contributions and convincingly demonstrate the value of the proposed hierarchical approach. The AC is very familiar with this area and also thoroughly checked the submission as well as the discussion.

The author response and updates to the manuscript greatly improve the quality of the contributions and clarify novelty compared to ComENet. We also encourage the authors to incorporate their new results on four Atom3D additional protein tasks (RES, MSP, PSR, LEP) into the appendix as they suggested.

**Note From Pc:**

if the above contains the word "oral" or "spotlight" please see: "oral" presentation means -> notable-top-5% and "spotlight" means -> notable-top-25%. As stated in our emails, we are disassociating presentation type from AC recommendations